# Satellite glia modulate sympathetic neuron survival, activity, and autonomic function

Aurelia A Mapps[1], Erica Boehm[1], Corinne Beier[2], William T Keenan[1†], Jennifer Langel[2], Michael Liu[1], Michael B Thomsen[2], Samer Hattar[2], Haiqing Zhao[1], Emmanouil Tampakakis[3], Rejji Kuruvilla[1]*

[1]Department of Biology, Johns Hopkins University, Baltimore, United States; [2]Section on Light and Circadian Rhythms (SLCR), National Institute of Mental Health, Bethesda, United States; [3]Department of Medicine, Division of Cardiology, Johns Hopkins University, Baltimore, United States

*For correspondence:
rkuruvilla@jhu.edu

Present address: †Department of Neuroscience, The Scripps Research Institute, California, United States

Competing interest: The authors declare that no competing interests exist.

**Abstract** Satellite glia are the major glial cells in sympathetic ganglia, enveloping neuronal cell bodies. Despite this intimate association, the extent to which sympathetic functions are influenced by satellite glia in vivo remains unclear. Here, we show that satellite glia are critical for metabolism, survival, and activity of sympathetic neurons and modulate autonomic behaviors in mice. Adult ablation of satellite glia results in impaired mTOR signaling, soma atrophy, reduced noradrenergic enzymes, and loss of sympathetic neurons. However, persisting neurons have elevated activity, and satellite glia-ablated mice show increased pupil dilation and heart rate, indicative of enhanced sympathetic tone. Satellite glia-specific deletion of Kir4.1, an inward-rectifying potassium channel, largely recapitulates the cellular defects observed in glia-ablated mice, suggesting that satellite glia act in part via K+-dependent mechanisms. These findings highlight neuron–satellite glia as functional units in regulating sympathetic output, with implications for disorders linked to sympathetic hyper-activity such as cardiovascular disease and hypertension.

## Editor's evaluation

This study covers the understudied topic of satellite glia and its impact on sympathetic neuron survival. An unexpected increase in neuronal activity was observed when satellite glia were deleted by mouse genetics. Novel roles of satellite glia in sympathetic physiology were uncovered by physiology assays. An extensive response to the reviewers, along with the addition of supporting data, was provided by the authors.

## Introduction

The sympathetic nervous system prepares the body for 'fight or flight' responses and maintains homeostasis during daily activities such as exercise, digestion, or regulation of body temperature. Post-ganglionic neurons, which reside in sympathetic ganglia and project axons to innervate diverse peripheral organs and tissues, mediate key autonomic effects, including cardiac output, metabolism, and immune function (*Goldstein, 2013*). Satellite glia are the major glial cells in sympathetic ganglia (*Hanani, 2010*; *Mapps et al., 2021*) and have a unique architecture in completely enveloping neuronal cell bodies (*Elfvin and Forsman, 1978*; *Gabella et al., 1988*). Each neuron and associated glia are thought to form a discrete structural and functional unit (*Hanani, 2010*). Despite this intimate

association, the functions of satellite glial cells in the sympathetic nervous system, particularly in vivo, are vastly understudied.

Satellite glia have been largely characterized by their distinctive location and morphology in peripheral ganglia (*Elfvin and Forsman, 1978*; *Gabella et al., 1988*). Like neurons, satellite glial cells are derived from multipotent neural crest precursors and form thin cytoplasmic sheaths around cell bodies, dendrites, and synapses of sympathetic neurons, with only 20 nm of space, the width of a synaptic cleft, between neuronal and glial membranes (*Elfvin and Forsman, 1978*; *Gabella et al., 1988*; *Hanani, 2010*; *Pannese, 1981*). Multiple satellite glia surround a single neuron and are connected with each other, and with neurons, via gap junctions, with the number of glial cells per neuron being positively correlated to soma size (*Elfvin and Forsman, 1978*; *Hanani, 2010*; *Ledda et al., 2004*). This unique arrangement places satellite glial cells in an ideal position to be critical regulators of neuronal connectivity, synaptic transmission, and homeostasis. Studies in sympathetic neuron–glia co-cultures have suggested roles for satellite glia in promoting dendrite growth, synapse formation, modulating extracellular ion and neurotransmitter concentrations, and regulating synaptic transmission (*Enes et al., 2020*; *Feldman-Goriachnik et al., 2018*; *Hanani, 2010*; *Tropea et al., 1988*). Satellite glia also envelop neuronal cell bodies in sensory ganglia in the peripheral nervous system (PNS) (*Hanani and Spray, 2020*). Recent studies implicate sensory satellite glia in regulating chronic pain through modulating neuronal hyper-excitability (*Kim et al., 2016*), and in promoting axon regeneration after peripheral nerve injury in vivo (*Avraham et al., 2020*). Satellite glia have been proposed to be closest to astrocytes in the central nervous system (CNS) with respect to expression of machinery related to neurotransmitter uptake/turnover, inward-rectifying potassium channels, functional coupling via gap junctions, and close association with synapses (*Hanani and Spray, 2020*; *Hanani and Verkhratsky, 2021*). In contrast to the wealth of information on CNS astrocytes and emerging evidence of the significance of sensory satellite glia in the PNS, little is known about the functions of satellite glia in the sympathetic nervous system in vivo.

Here, using genetic ablation in mice, we reveal that loss of satellite glia results in impaired metabolic signaling, soma atrophy, reduced expression of noradrenergic enzymes, and enhanced apoptosis of adult sympathetic neurons. The persisting neurons, however, show elevated activity as revealed by increased neuronal c-Fos expression. Consistently, satellite glia-ablated mice had enhanced circulating norepinephrine (NE) and elevated heart rate, indicative of heightened sympathetic tone. We further deleted Kir4.1, an inward-rectifying potassium channel, specifically in satellite glia in mice. Satellite glia-specific deletion of Kir4.1 largely recapitulates the cellular phenotypes observed in glia-ablated mice, suggesting that satellite glia support neurons, in part, via $K^+$-dependent mechanisms. These findings reveal that satellite glia provide critical metabolic and trophic support to sympathetic neurons and are modulators of the ganglionic milieu, neuronal activity, and resulting autonomic behaviors.

## Results

### Inducible ablation of satellite glia using BLBP:iDTA mice

*Fabp7*, which encodes for brain lipid binding protein (BLBP), a fatty acid transporter, is one of the most highly expressed transcripts in mouse satellite glial cells (*Avraham et al., 2020*; *Kurtz et al., 1994*; *Mapps et al., 2022*). Using immunostaining for BLBP and tyrosine hydroxylase (TH), a marker for noradrenergic sympathetic neurons, we observed BLBP-positive satellite glia in the mouse sympathetic ganglia (superior cervical ganglia [SCG]) at both developmental and adult stages (*Figure 1A*). BLBP labeling was 'patchy' in the embryonic and neonatal ganglia, a period when satellite glia undergo migration into the ganglia and are also proliferating (*Hall and Landis, 1992*). However, by 2 weeks after birth, BLBP-positive satellite glial cells had expanded processes around neuronal cell bodies and formed thin ring-like glial sheaths around neuronal somas when observed at P31 (*Figure 1A*).

In the adult PNS, BLBP expression is restricted to satellite glia and is not detected in Schwann cells based on single-cell RNA-sequencing analysis and characterization of *Fabp7-CreER2*-driven reporter mice (*Avraham et al., 2020*). Using data sets from our recently published single-cell RNA-sequencing analysis of peripheral sympathetic and sensory ganglia from adult mice (postnatal days 30–45) (*Mapps et al., 2022*), we found that BLBP (*Fabp7*) is ~140-fold enriched in satellite glial cells compared to other ganglionic cell types (*Figure 1—figure supplement 1A*). Of note, Schwann cells are scarce in adult sympathetic ganglia compared to sensory ganglia based on single-cell RNA-sequencing analyses

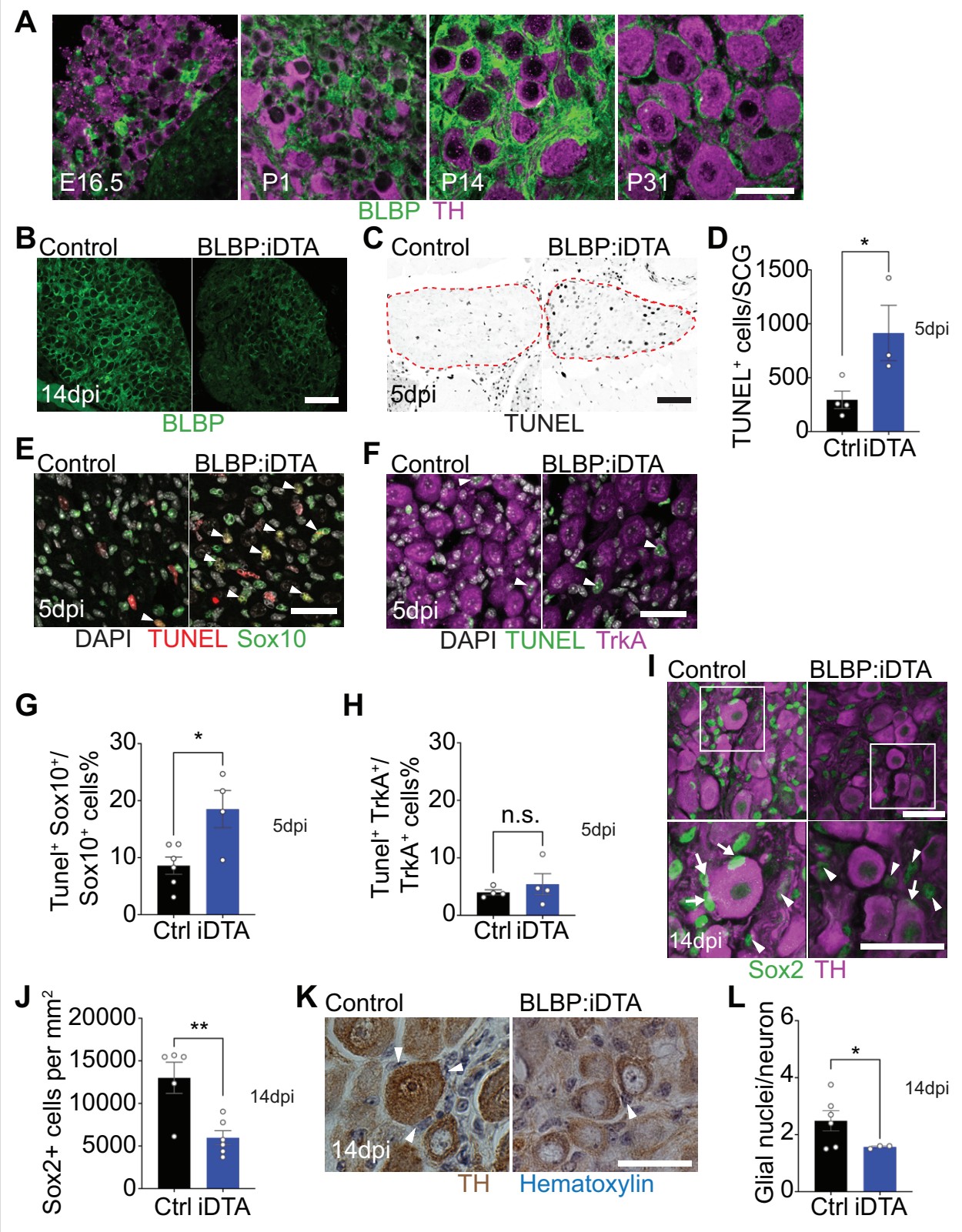

**Figure 1.** DTA-mediated ablation of satellite glia in sympathetic ganglia. (**A**) Satellite glial cells, immunolabeled with brain lipid binding protein (BLBP, green), progressively ensheathe sympathetic neuron cell bodies, labeled with tyrosine hydroxylase (TH, magenta) in the superior cervical ganglia (SCG) during development. Time points shown are embryonic day 16.5 (E16.5) and postnatal days P1, P14, and P31. Scale bar: 30 μm. (**B**) Reduced BLBP expression in BLBP:iDTA SCG relative to control ganglia at 14 days after the last tamoxifen injection (14 dpi). Scale bar: 100 μm. (**C**) Increased apoptosis

*Figure 1 continued on next page*

*Figure 1 continued*

in BLBP:iDTA SCG (outlined in red dashed line) compared to controls as detected by TUNEL labeling at 5 days post-tamoxifen injection (5 dpi). Scale bar: 100 µm. (**D**) Quantification of apoptotic cells in SCGs at 5 dpi from n = 4 control and 3 mutant mice. Data are means ± SEM. *p<0.05, *t*-test. (**E**) Increased apoptosis of satellite glial cells assessed by Sox10 immunostaining (green) and TUNEL labeling (red) in BLBP:iDTA sympathetic ganglia compared to control at 5 dpi. DAPI channel is shown in gray. Arrowheads indicate TUNEL$^+$;Sox10$^+$ cells. Scale bar: 30 µm. (**F**) Neuronal apoptosis is similar between BLBP:iDTA and control ganglia at 5 dpi as assessed by TrkA immunostaining (magenta) and TUNEL labeling (green). DAPI channel is shown in gray. Arrowheads indicate TUNEL$^+$;TrkA$^+$ neurons. Scale bar: 30 µm. (**G**) Quantification of TUNEL$^+$;Sox10$^+$ cells expressed as a % of Sox10$^+$ cells. Data are presented as means ± SEM from n = 6 control and 4 mutant mice, **p<0.05, *t*-test. (**H**) Quantification of TUNEL$^+$;TrkA$^+$ cells expressed as a % of TrkA$^+$ cells. Data are presented as means ± SEM from n = 4 mice per genotype, n.s, not significant, *t*-test. (**I**) Decreased association of Sox2-labeled satellite glia (green) with TH-positive sympathetic neurons (magenta) in BLBP:iDTA sympathetic ganglia compared to controls at 14 dpi. Arrows indicate Sox2-labeled nuclei of satellite glia associated with TH-positive sympathetic neuron cell bodies, while arrowheads indicate Sox2-labeled nuclei not abutting neuronal soma. Gain in both images has been set to the same level for a better visualization of Sox2-labeled nuclei in mutant ganglia. Scale bar: 30 µm. (**J**) Quantification of Sox2-positive cells in control and BLBP:iDTA SCGs at 14 dpi. Data are presented as mean ± SEM from n = 5 control and 6 mutant animals, **p<0.01, *t*-test. (**K**) TH DAB immunohistochemistry and hematoxylin staining shows fewer satellite glia nuclei (blue) associated with TH-labeled sympathetic neuron cell bodies (brown) in BLBP:iDTA ganglia at 14 dpi. Scale bar: 30 µm. (**L**) Quantification of glial nuclei associated with neuronal soma. Data are presented as means ± SEM from n = 6 control and 3 mutant animals, **p<0.05, *t*-test.

The online version of this article includes the following source data and figure supplement(s) for figure 1:

**Source data 1.** Raw data for neuronal and glia apoptosis and glia cell counts.

**Figure supplement 1.** Brain lipid binding protein (BLBP) is a specific marker for adult satellite glial cells.

**Figure supplement 1—source data 1.** Raw data for normalized Fabp7 in sympathetic ganglia.

**Figure supplement 2.** Satellite glia ablation in BLBP:iDTA sympathetic ganglia.

**Figure supplement 2—source data 1.** Raw data for glia cell counts and transcript changes in sympathetic ganglia.

**Figure supplement 3.** Satellite glia depletion does not induce macrophage infiltration or proliferative changes in sympathetic ganglia.

**Figure supplement 3—source data 1.** Raw data for macrophage and cell proliferation analyses.

(*Mapps et al., 2022*). To further ensure the cellular specificity of BLBP, we generated genetic reporter mice by crossing *Fabp7-CreER2* mice with *ROSA26$^{mEGFP}$* mice (*Muzumdar et al., 2007*), which drives expression of membrane-tagged EGFP (*Figure 1—figure supplement 1B*). We found that tamoxifen treatment (180 mg/kg body weight for five consecutive days) in *Fabp7-CreER2;ROSA26$^{mEGFP}$* mice resulted in m-EGFP reporter expression in satellite glia marked by Kir4.1 immunolabeling (*Vit et al., 2006*), but not in TH-positive sympathetic neurons, IBA1-labeled macrophages, or Pdgfrβ-labeled vascular mural cells in sympathetic ganglia (*Figure 1—figure supplement 1C–F*). Together, these results are consistent with previous studies (*Avraham et al., 2022*; *Avraham et al., 2020*; *Avraham et al., 2021*; *Mapps et al., 2022*) showing that BLBP is a specific marker for satellite glia in adult mice.

Next, to accomplish ablation of satellite glia, we crossed *Fabp7-CreER2* mice (*Maruoka et al., 2011*) with *ROSA26$^{eGFP-DTA}$* mice (*Ivanova et al., 2005*), where Cre drives expression of a copy of diphtheria toxin subunit A (DTA). At postnatal day 30, *Fabp7-CreER2;ROSA26$^{eGFP-DTA}$* mice were injected with either vehicle (corn oil) or tamoxifen (180 mg/kg body weight) for five consecutive days, and all analyses were performed at 5 or 14 days post injection. By 2 weeks, we observed a drastic loss of BLBP expression in sympathetic (superior cervical) ganglia from tamoxifen-treated mice (henceforth referred to as BLBP:iDTA mice) compared to vehicle-injected control (Ctrl) mice (*Figure 1B*). While satellite glia in control ganglia formed characteristic ring-like structures around neuronal soma, glial organization was disrupted in mutant mice with diminished BLBP staining detected within the ganglia. To determine whether DTA expression results in cell death, we assessed apoptosis using TUNEL labeling and observed a threefold increase in TUNEL-positive cells in BLBP:iDTA sympathetic ganglia compared to controls at 5 days post-tamoxifen injection (*Figure 1C and D*). To identify cell types undergoing apoptosis, we performed TUNEL labeling together with immunostaining for Sox10, a transcription factor expressed in satellite glia (*Britsch et al., 2001*), and TrkA, a sympathetic neuron marker (*Fagan et al., 1996*). We found a twofold increase in the number of TUNEL$^+$;Sox10$^+$ cells in BLBP:iDTA ganglia compared to controls at 5 days post-tamoxifen injection (5 dpi) (*Figure 1E–G*). Although there was a trend toward enhanced neuronal apoptosis, the number of TUNEL$^+$;TrkA$^+$ sympathetic neurons in BLBP:iDTA ganglia was not statistically different from that in controls at this stage (*Figure 1F–H*). Despite increased glial apoptosis at 5 dpi, there was no significant loss of satellite glial cells at this time as assessed by quantification of Sox2-immunoreactive cells (*Figure 1—figure supplement 2A and B*), where Sox2 is another transcription factor expressed in satellite glia (*Koike*

et al., 2014). However, at 5 dpi, we did observe fewer Sox2-positive glial cells juxtaposed to cell bodies of individual sympathetic neurons in mutant ganglia (*Figure 1—figure supplement 2 A and C*), suggesting that unhealthy/dying satellite glia lose their contacts with neuronal soma. By 14 days post-tamoxifen injection (14 dpi), however, there was a pronounced loss of Sox2-labeled satellite glial cells (54.2% decrease) in sympathetic ganglia from BLBP:iDTA mice (*Figure 1I and J*). To confirm ablation of Sox2-positive cells, rather than downregulated Sox2 expression in mutant ganglia, we generated binary images of Sox2-labeled cells by filtering and thresholding using ImageJ. This method allowed us to simply record the presence or absence of cells in the images in a manner independent of pixel values. Quantification of binary images revealed a substantial decrease in the number of Sox2-labeled cells (33% decrease) in BLBP:iDTA ganglia compared to controls at 14 dpi (*Figure 1—figure supplement 2D and F*). Of note, this 33% decrease is lower than the 54% loss quantified using Sox2 immunofluorescence (*Figure 1J*). Assessing binary images from the 5 dpi time point indicated that there was no significant loss of Sox2-labeled cells at this stage (*Figure 1—figure supplement 2E and G*), consistent with quantifications based on Sox2-immunofluorescence (see *Figure 1—figure supplement 2B*). As an additional measure to visualize sympathetic neurons and associated satellite glial cells in a manner that does not rely on fluorescence, we labeled sympathetic neurons using TH DAB (3,3'-diaminobenzidine) immunohistochemistry and visualized satellite glia nuclei abutting TH-positive neuronal cell bodies based on their distinctive appearance and location using hematoxylin staining. We observed a significant loss of glial nuclei associated with soma of individual sympathetic neurons in BLBP:iDTA mice compared to control mice at 14 dpi (*Figure 1K and L*). Together, these results indicate that satellite glia are undergoing apoptosis by 5 dpi, and by 14 dpi, there is a substantial loss of glial cells in BLBP:iDTA ganglia. Further, consistent with cell loss at 14 dpi, we found a pronounced downregulation of several satellite glia-specific transcripts, including *Fabp7*, *Fasn*, *Apoe*, and *Kcnj10* (*Mapps et al., 2022*), in BLBP:iDTA sympathetic ganglia at this time as revealed by quantitative PCR (qPCR) analyses (*Figure 1—figure supplement 2H*).

In BLBP:iDTA ganglia, we did not find increased macrophage infiltration as assessed by quantification of IBA-1-labeled cells (*Figure 1—figure supplement 3A and B*). It remains to be determined whether satellite glia loss triggers increased macrophage reactivity or phagocytosis. Further, EdU labeling indicated that satellite glial cells do not undergo increased proliferation as a compensatory or injury-induced response after DTA expression (*Figure 1—figure supplement 3C and D*). Together, these results suggest that DTA-induced ablation of satellite glia is not accompanied by increased macrophage density or glial proliferation in mutant sympathetic ganglia.

## Satellite glia depletion impairs noradrenergic enzyme expression, metabolism, and survival of sympathetic neurons

When performing TH immunostaining to visualize sympathetic neuron morphology, we noticed a striking downregulation of TH immunoreactivity in sympathetic neuronal cell bodies from BLBP:iDTA mice. TH is the rate-limiting enzyme in the biosynthesis of NE, the classical sympathetic neurotransmitter. qPCR analyses revealed a drastic downregulation in *Th* and *Dopamine Beta-Hydroxylase* (DBH) transcript levels, (88 and 99% decrease, respectively) (*Figure 2A and B*). DBH converts dopamine to NE in the NE biosynthetic pathway. Thus, satellite glia maintain noradrenergic enzymes in sympathetic neurons.

Using TH immunohistochemistry, we also observed pronounced atrophy of neuronal cell bodies in sympathetic ganglia after satellite glia loss (*Figure 2A*). Quantification of soma sizes revealed significantly reduced neuronal soma areas in satellite glia-depleted mice, with a greater distribution of smaller-sized cell bodies in mutant ganglia relative to controls (*Figure 2C*). Median values for soma areas are $162 \pm 17 \ \mu m^2$ for control neurons vs $118 \pm 7.1 \ \mu m^2$ for mutant neurons (*Figure 2D*). Since the PI3K/Akt/mTOR pathway is a known regulator of neuronal soma size (*Kwon et al., 2006*; *van Diepen et al., 2009*; *Zhou et al., 2009*), we assessed expression of key downstream effectors of the mTOR pathway, phosphorylated ribosomal protein 6 (p-S6) and phosphorylated eukaryotic translation initiation factor 4E (eIF4E)-binding protein 1 (p-4-EBP1) (*Meyuhas, 2008*; *Saxton and Sabatini, 2017*), using immunostaining. We observed a dramatic reduction in p-S6 and p-4-EBP1 immunoreactivity in satellite glia-depleted ganglia (*Figure 2E*, *Figure 2—figure supplement 1A*). These results suggest that satellite glia promote soma growth, in part, via regulating mTOR signaling in sympathetic neurons.

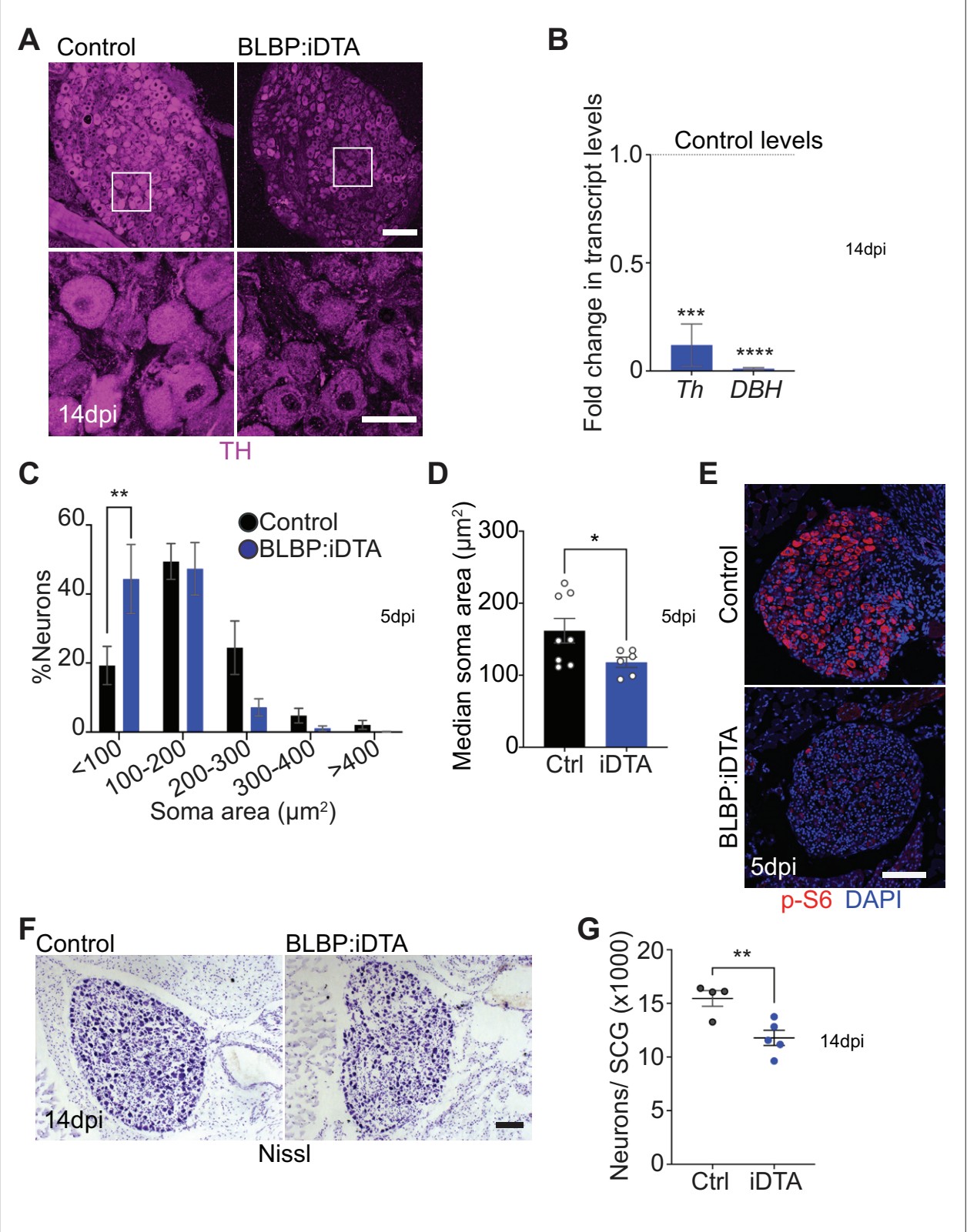

**Figure 2.** Neuronal defects in norepinephrine (NE) biosynthesis, metabolism, and survival in satellite glia-depleted mice. (**A**) Tyrosine hydroxylase (TH) expression is downregulated in BLBP:iDTA sympathetic neurons. Insets also show atrophied neuronal cell bodies in mutant ganglia compared to controls. Scale bar: 100 µm for upper panels and 30 µm for insets. (**B**) Transcripts for *Th* and *DBH*, key enzymes in norepinephrine biosynthesis, are decreased in BLBP:iDTA SCG relative to control ganglia. Data are presented as means ± SEM from superior cervical ganglia (SCG) collected from n =

*Figure 2 continued on next page*

*Figure 2 continued*

3–4 animals per genotype. ***p<0.001, ****p<0.0001, *t*-test with Bonferroni–Dunn's correction. (**C**) Histogram shows a greater distribution of smaller soma sizes in mutant neurons compared to controls. Results are means ± SEM from n = 6–8 animals per genotype, **p<0.01, two-way ANOVA with Bonferroni's correction. (**D**) Reduced soma sizes, represented as median values for soma areas (μm$^2$), of sympathetic neurons from BLBP:iDTA mice compared to controls at 5 dpi. Values are means ± SEM from n = 8 control and 6 mutant animals, *p<0.05, *t*-test. (**E**) Immunostaining shows reduced p-S6 levels in BLBP:iDTA ganglia. Scale bar:100 μm. (**F, G**) Cell counts in Nissl-stained SCG tissue sections show reduced sympathetic neuron numbers in satellite glia-depleted mice 14 dpi. Results are means ± SEM from n = 4 control and 5 mutant animals. **p<0.01, *t*-test.

The online version of this article includes the following source data and figure supplement(s) for figure 2:

**Source data 1.** Raw data for neuronal morphology and survival.

**Figure supplement 1.** Neuronal morphology and signaling in BLBP:iDTA mice.

**Figure supplement 1—source data 1.** Raw data for neuronal morphology and survival.

Given the soma atrophy, we asked whether satellite glia ablation affects neuron survival. At 5 dpi, sympathetic neuron numbers in BLBP:iDTA mice were not significantly different from that in control littermates (12,240 ± 1514 in mutant vs. 14,200 ± 1005 in control mice) as revealed by Nissl staining and quantification of cell counts in tissue sections from SCG (*Figure 2—figure supplement 1B*). However, by 14 dpi, we found a marked loss of sympathetic neurons (24% decrease) in satellite glia-ablated ganglia (11,790 ± 706 mutant neurons vs. 15,460 ± 734 control neurons) (*Figure 2F and G*). Despite the decrease in neuronal numbers, sympathetic axon innervation was maintained in target organs in BLBP:iDTA mice, when assessed by wholemount TH immunostaining in iDISCO-cleared tissues and light sheet microscopy (*Figure 2—figure supplement 1C–E*). Intriguingly, TH levels in mutant axons appeared to be similar to that in controls (*Figure 2—figure supplement 1C–E*), despite the marked reduction in *Th* mRNA and protein in neuronal cell bodies residing in the ganglia (see *Figure 2A and B*), suggesting differential regulation of TH distribution in neuronal soma *vs.* axons.

Together, these results indicate that satellite glia provide trophic and metabolic support to adult sympathetic neurons and regulate noradrenergic biosynthetic machinery, specifically in neuronal cell bodies.

## Sympathetic activity is elevated in satellite glia-depleted mice

Given neuronal deficits with satellite glia depletion, we sought to determine whether autonomic responses were impacted in BLBP:iDTA mice. In mammals, pupil size can serve as a noninvasive and rapid readout for autonomic function (*McDougal and Gamlin, 2015*). Pupil size is modulated by a balance of sympathetic vs. parasympathetic activity, with the sympathetic component regulating pupil dilation while parasympathetic activity controls pupil constriction (*McDougal and Gamlin, 2015*). To measure basal pupil size, control and BLBP:iDTA mice were dark-adapted for 2 days, and pupil sizes recorded for 5–10 s in the dark in non-anesthetized mice (*Keenan et al., 2016*). Surprisingly, despite the neuronal loss and downregulated noradrenergic biosynthetic enzymes in neuronal soma, we observed increased basal pupil areas in BLBP:iDTA mice compared to controls (*Figure 3A and B*). To ask whether this phenotype is due to decreased parasympathetic activity, we measured pupil constriction in response to increasing light intensities, ranging from 0.01 to 1000 lux, administered for 30 s. Light onset at 0.1 lux or higher resulted in rapid constriction with greater constrictions at higher light intensities in both BLBP:iDTA and control mice (*Figure 3—figure supplement 1A*). The intensity responses were virtually identical for the two groups (*Figure 3—figure supplement 1A*). These results suggest that parasympathetic function is intact in BLBP:iDTA mice and that the enlarged pupil areas likely reflect an increase in sympathetic tone with the loss of satellite glia.

As a second and independent assessment of autonomic function, we measured heart rate and heart rate variability (HRV) using electrocardiogram (ECG) recordings in mice (*Thireau et al., 2008*). Increased sympathetic activity results in an accelerated heart rate and decreased HRV, defined as the variation in time intervals between consecutive heartbeats (*Thireau et al., 2008*). Strikingly, BLBP:iDTA mice exhibited increased heart rates (*Figure 3C and D*) and decreased HRV (*Figure 3—figure supplement 1B*) compared to controls. Tamoxifen injections had no effect on heart rate in wild-type C57BL/6J mice or control *ROSA26*$^{eGFP-DTA}$ mice that do not express Cre (*Figure 3—figure supplement 1C and D*).

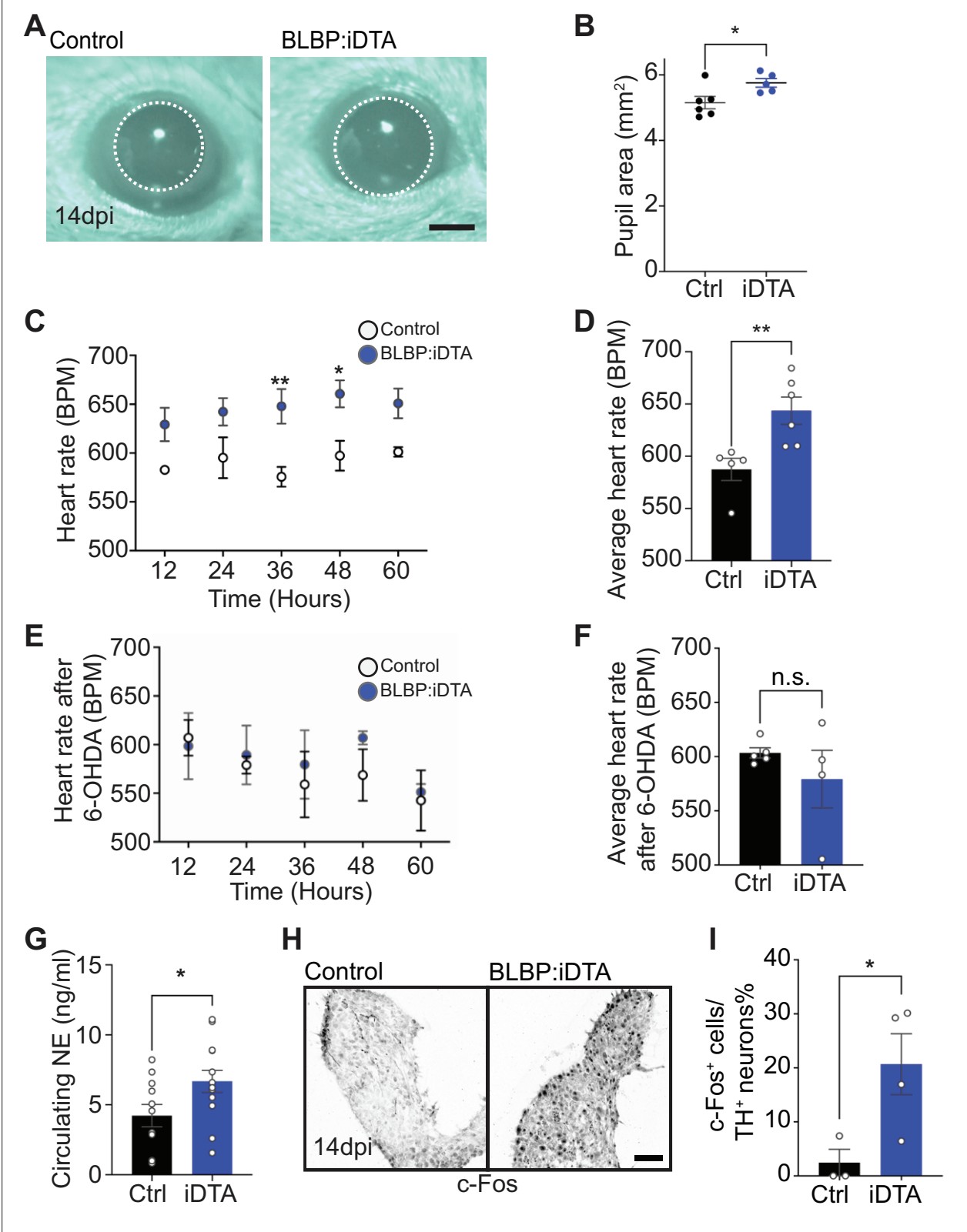

**Figure 3.** Elevated sympathetic activity in satellite glia-depleted mice. (**A, B**) Dark-adapted BLBP:iDTA mice have increased basal pupil size compared to control littermates. Results are presented as mean ± SEM from n = 6 control and 5 mutant animals. *p<0.05, *t*-test. (**C**) BLBP:iDTA mice exhibit elevated heart rate, relative to controls. ECGs were recorded continuously in conscious mice for 7 days, although only data for fourth to seventh days after insertion of lead implants are included in the analysis. Results are presented as mean ± SEM from n = 5 control and 6 mutant animals. *p<0.05,

*Figure 3 continued*

**p<0.01, two-way ANOVA with Bonferroni's correction. (**D**) Average heart rate over fourth to seventh days after lead implantation. Results are mean ± SEM from n = 5 control and 6 mutant animals. **p<0.01, *t*-test. (**E, F**) 6-Hydroxydopamine (6-OHDA) administration (150 mg/kg, i.p.) prevents elevated heart rate in BLBP:iDTA mice. Results are mean ± SEM from n = 5 control and 4 mutant animals, n.s, not significant, *t*-test. (**G**) Increased circulating norepinephrine levels in BLBP:iDTA mice. Results are mean ± SEM from n = 11 control and 13 mutant animals. *p<0.05, *t*-test. (**H, I**) Increased c-Fos-positive sympathetic neurons in mutant ganglia. Quantification of c-Fos$^+$;TH$^+$ sympathetic neurons as a % of total number of TH$^+$ sympathetic neurons. Results are mean ± SEM from n = 3 control and 4 mutant animals. *p<0.05, *t*-test.

The online version of this article includes the following source data and figure supplement(s) for figure 3:

**Source data 1.** Raw data for pupil size and heart rate in mice, NE secretion, and c-Fos expression.

**Figure supplement 1.** Autonomic function analyses in BLBP:iDTA mice.

**Figure supplement 1—source data 1.** Raw data for pupil and heart rate analyses in mice and adrenergic receptor expression in heart tissue.

To directly test the involvement of sympathetic neurons in the heart defect observed in BLBP:iDTA mice, we chemically ablated sympathetic nerves using 6-hydroxydopamine (6-OHDA), which selectively destroys sympathetic, but not parasympathetic or sensory nerves (*Borden et al., 2013*; *Kostrzewa and Jacobowitz, 1974*). Further, 6-OHDA does not cross the blood–brain barrier after intraperitoneal injections in adult animals, restricting its actions to the periphery (*Kostrzewa and Jacobowitz, 1974*). Ablation of sympathetic nerves prevented the augmented heart rate in BLBP:iDTA mice (*Figure 3E and F*), suggesting the dysfunction of peripheral sympathetic neurons as the primary contributor to the cardiac defect in satellite glia-ablated mice.

To understand the molecular and/or cellular basis for enhanced sympathetic activity in satellite glia-depleted mice, we measured circulating NE levels and observed a significant 1.6-fold increase in BLBP:iDTA mice compared to controls (*Figure 3G*). These results suggest that, despite the down-regulation of NE biosynthetic machinery in neuronal cell bodies (see *Figure 2A and B*), neurotransmitter secretion is augmented in sympathetic axons and/or its reuptake/degradation decreased, in satellite glia-depleted mice. NE acts through α- and β-adrenergic receptors in target tissues. Heightened sympathetic activity and increased circulating NE elicits the compensatory downregulation of adrenergic receptor levels and activities in target tissues (*Eschenhagen, 2008*). We found significantly decreased expression of *Adrb1*, the major adrenergic receptor for NE signaling in the heart (*de Lucia et al., 2018*), as well as *Adrb2* and *Adra2c*, using qPCR analyses of cardiac tissue (*Figure 3—figure supplement 1E*). Lastly, immunostaining for c-Fos, an immediate early transcription factor that serves as a reporter of neuronal activity (*Sheng and Greenberg, 1990*), revealed a striking eightfold increase in the number of c-Fos-positive sympathetic neurons in BLBP:iDTA ganglia compared to controls (*Figure 3H and I*).

Together, these results indicate that satellite glia depletion results in elevated sympathetic neuron activity, impaired NE homeostasis, and autonomic behavioral defects in mice.

## Satellite glia-specific deletion of *Kir4.1* disrupts sympathetic neuron activity

Kir4.1 is a glial-specific, inwardly rectifying K$^+$ channel that shows the highest expression in astrocytes and satellite glia (*Hanani and Spray, 2020*; *Olsen et al., 2015*; *Vit et al., 2008*). *Kcnj10* mRNA for Kir4.1 is highly enriched in satellite glia (approximately threefold enriched) compared to other ganglionic cell types based on our analysis of data sets obtained from single-cell RNA-sequencing of sympathetic and sensory ganglia from adult mice (postnatal days 30–45) (*Mapps et al., 2022*; *Figure 4—figure supplement 1A*). Co-immunolabeling for Kir4.1 and m-EGFP in *Fabp7-CreER2;ROSA26$^{mEGFP}$* reporter mice revealed co-localization of the two signals in sympathetic ganglia (*Figure 1—figure supplement 1C*). Further, in BLBP:iDTA sympathetic ganglia, we observed a pronounced decrease in *Kcnj10* mRNA compared to controls as assessed by qPCR analysis (*Figure 1—figure supplement 2H*) and single-molecule fluorescence in situ hybridization (smFISH) (*Figure 4—figure supplement 1B*). Our results are consistent with previous studies that *Kcnj10* is enriched in satellite glial cells (*Avraham et al., 2020*; *Hanani and Spray, 2020*; *Vit et al., 2008*).

In mice, global or tissue-specific Kir4.1 deletion results in impaired K$^+$ and glutamate homeostasis, loss of glial K$^+$ conductance, neuronal excitability, epileptic seizures, and pain-like behaviors (*Cui et al., 2018*; *Djukic et al., 2007*; *Olsen et al., 2015*; *Tang et al., 2010*; *Vit et al., 2008*). To

ask whether impaired Kir4.1 function might contribute to the neuronal defects observed in satellite glia-ablated mice, we generated satellite glia-specific Kir4.1 knockout mice (Kir4.1 cKO) by crossing *Fabp7-CreER2* mice to *Kcnj10* floxed mice (*Djukic et al., 2007*). *Fabp7-CreER2; Kcnj10*<sup>fl/fl</sup> (henceforth called Kir4.1 cKO) mice were treated with vehicle or tamoxifen for five consecutive days to conditionally delete Kir4.1 from satellite glial cells. Expression of Kir4.1, visualized by immunofluorescence, was significantly reduced in Kir4.1 cKO sympathetic ganglia (*Figure 4A*), and qPCR analysis indicated a 76% decrease in *Kcnj10* transcript levels (*Figure 4B*).

Loss of Kir4.1 did not alter satellite glial numbers in sympathetic ganglia as assessed by quantification of Sox2-positive cells (*Figure 4C and D*, *Figure 4—figure supplement 1C and D*). Kir4.1 deletion also did not affect *Fabp7* (BLBP) expression in satellite glia assessed by qPCR analysis and immunostaining (*Figure 4—figure supplement 1E and F*). BLBP immunostaining also indicated that the characteristic ring-like glial organization around neuronal soma was not perturbed in Kir4.1 cKO sympathetic ganglia (*Figure 4—figure supplement 1F*). However, we observed a marked increase in the number of c-Fos-positive cells in Kir4.1 cKO SCGs compared to controls (*Figure 4E and F*). The majority of c-Fos-positive cells in Kir4.1 cKO ganglia appeared to be neurons based on their morphology. Intriguingly, despite increased c-Fos-positive sympathetic neurons, Kir4.1 deletion resulted in downregulated expression of noradrenergic biosynthetic enzymes, TH and DBH, in neuronal cell bodies (*Figure 4C, G, and H*), similar to the phenotype observed with satellite glia depletion.

Together, these results suggest that satellite glial cells control sympathetic neuron activity and noradrenergic enzyme expression via Kir4.1-dependent regulation of the neuronal microenvironment.

## Satellite glia-specific deletion of Kir4.1 elicits neuron atrophy and apoptosis

We next addressed the relevance of satellite glia Kir4.1 expression in sympathetic neuron viability. Quantification of soma sizes revealed that neurons undergo atrophy in Kir4.1 cKO sympathetic ganglia (*Figure 5A*), similar to the defect observed with DTA-induced loss of satellite glia. Soma areas from tissue sections, represented as median values, were $155 \pm 13 \ \mu m^2$ for control neurons vs. $118 \pm 3.6 \ \mu m^2$ for mutant neurons (*Figure 5B*). Kir4.1 loss in satellite glial cells also elicited a pronounced decrease in the levels of p-S6 and p-4-EBP1, two well-established downstream effectors of mTOR activity (*Meyuhas, 2008*; *Saxton and Sabatini, 2017*), in sympathetic neurons (*Figure 5C*, *Figure 5—figure supplement 1A*), similar to the effects of satellite glia ablation (*Figure 2E*, *Figure 2—figure supplement 1A*). Notably, we observed an eightfold increase in apoptotic cells in Kir4.1 cKO sympathetic ganglia using TUNEL labeling at 14 days post-tamoxifen injection (*Figure 5D and E*). The apoptotic cells were primarily sympathetic neurons based on co-labeling for TUNEL and TrkA (*Figure 5—figure supplement 1B and C*). We did not observe increased apoptosis of satellite glial cells in Kir4.1 cKO sympathetic ganglia compared to controls (*Figure 5—figure supplement 1D and E*), consistent with results that the number of Sox2-positive satellite glia was not altered by Kir4.1 deletion (see *Figure 4C and D*). Consistent with enhanced neuronal apoptosis, we observed a 22% decrease in sympathetic neuron numbers in Kir4.1 cKO sympathetic ganglia by 2 weeks after tamoxifen injections (*Figure 5F and G*). Similar to the findings in BLBP:iDTA mice, sympathetic axon innervation and axonal TH expression were similar between Kir4.1 cKO mice and controls (*Figure 5—figure supplement 1F and G*), despite the decrease in sympathetic neuron numbers and downregulated TH expression in neuronal cell bodies. Together, these results highlight that Kir4.1 is necessary for the survival of adult sympathetic neurons.

We next asked whether satellite glia-specific Kir4.1 deletion recapitulates the autonomic defects observed in BLBP:iDTA mice. Measurements of basal pupil size, heart rate, and circulating NE levels indicated that these parameters were not significantly altered in Kir4.1 cKO mice (*Figure 5H and I*, *Figure 5—figure supplement 1H–J*), despite increased neuronal activity as assessed by c-Fos immunoreactivity (see *Figure 4E and F*). Parasympathetic activity assessed by pupil constriction in response to different light intensities was also normal in Kir4.1 cKO mice (*Figure 5—figure supplement 1K*).

Together, these results suggest that Kir4.1 deletion recapitulates the cellular phenotypes observed in BLBP:iDTA mice, notably, increased neuron activity, impaired metabolic signaling, and enhanced apoptosis. However, the deletion of a single gene, *Kcnj10* encoding for Kir4.1, from satellite glial cells is not sufficient to drive behavioral changes at the whole animal level as seen with genetic ablation of satellite glial cells. These results suggest that there are other satellite glia-dependent mechanisms,

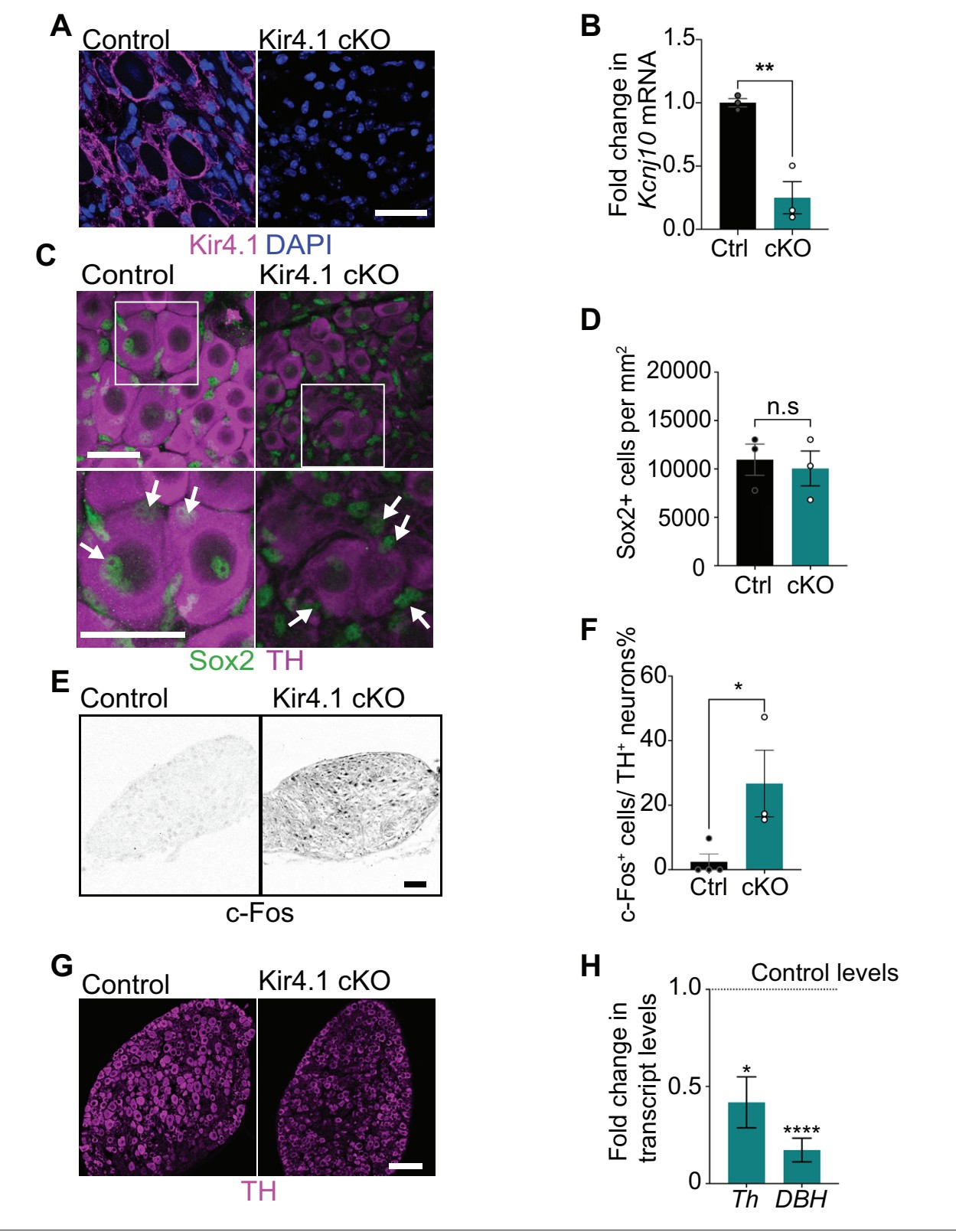

**Figure 4.** Satellite glia-specific Kir4.1 loss impairs norepinephrine (NE) enzyme expression and neuron activity. (**A, B**) Reduced Kir4.1 protein and transcript (*Kcnj10*) in Kir4.1 cKO mice. Scale bar: 30 µm. Data in (**B**) are mean ± SEM from n = 3 animals per genotype, **p<0.01, *t*-test. (**C, D**) Sox2-positive satellite glial cell numbers are unaffected in Kir4.1 cKO SCG. Arrows indicate Sox2-labeled nuclei of satellite glia associated with sympathetic neuron cell bodies. Arrowheads indicate Sox2-labeled satellite glia that are not adjacent to neuronal soma. Scale bar: 30 µm. Data are presented as

*Figure 4 continued on next page*

*Figure 4 continued*

mean ± SEM from n = 3 animals per genotype, n.s., not significant, *t*-test. (**E, F**) Kir4.1 deletion in satellite glia results in an increase in c-Fos-positive sympathetic neurons. Quantification of c-Fos$^+$;TH$^+$ sympathetic neurons as a % of total number of TH$^+$ sympathetic neurons. Scale bar: 100 μm. Quantifications are mean ± SEM from n = 3 animals per genotype, *p<0.05, *t*-test. (**G, H**) Downregulation of NE biosynthetic enzymes, TH and DBH, in Kir4.1 cKO sympathetic ganglia. Results are mean ± SEM from n = 3–5 animals per genotype, *p<0.05, ****p<0.0001, *t*-test with Bonferroni's correction.

The online version of this article includes the following source data and figure supplement(s) for figure 4:

**Source data 1.** Raw data for glia and neuron cell counts and gene expression changes in Kir4.1 mutant and control mice.

**Figure supplement 1.** Kir4.1 expression and analyses of satellite glia in *Kir4.1 cKO* mice.

**Figure supplement 1—source data 1.** Raw data for glia cell counts and gene expression changes in Kir4.1 mutant and control mice.

in addition to Kir4.1 activity, that contribute to regulation of neuronal excitability to drive circuit-level changes.

## Discussion

Despite decades of research on the sympathetic nervous system, satellite glia, the major glial cells in sympathetic ganglia, have remained an enigmatic component of the system. Given their specific contacts with neuronal cell bodies, satellite glia also provide the rare opportunity to study how glia support somatic compartments. Here, we show that satellite glia modulate sympathetic neuron metabolism, survival, neurotransmitter homeostasis, activity, and autonomic functions in adult mice. Together, our findings provide in vivo evidence that neurons and their surrounding glial covers are functional units in the regulation of sympathetic output.

We reveal that a key role for satellite glia is in restraining neuronal activity in mature sympathetic neurons. Depletion of satellite glia in adult mice amplifies neuronal activity, leading to increased circulating levels of NE and elevated sympathetic tone. Glia-ablated mice show enhanced pupil size and heart rate, demonstrating the necessity of these cells in the dynamic regulation of autonomic functions in conscious and freely moving animals. Our findings that satellite glia limit neuronal activity are similar to reported functions of other glial cells that encapsulate neuronal cell bodies, in particular, astrocytes (*Allen and Lyons, 2018*), microglia (*Badimon et al., 2020*; *Cserép et al., 2020*), and cortex glia in *Drosophila* (*Yadav et al., 2019*). Inhibitory effects of satellite glia might enable sympathetic neurons to respond to a wider range of input strengths and/or serve as a neuroprotective mechanism to limit neurotoxicity under conditions of stress or pathology.

How do satellite glia limit sympathetic neuron activity? The enhanced neuronal c-Fos signals in Kir4.1 cKO and satellite glia-ablated mice suggest that glial regulation of ion homeostasis, specifically, K$^+$ clearance, is a key mechanism that contributes to inhibition of neuronal activity. Electrophysiological analyses show that satellite glia have high K$^+$ conductance, which is almost exclusively dependent on Kir4.1 expression (*Tang et al., 2010*). Satellite glial cells are also coupled to one another through gap junctions (*Huang et al., 2005*; *Kim et al., 2016*). RNAi-mediated knockdown of Kir4.1 or Connexin 43, a gap junction protein, enhances excitability of sensory neurons and evokes nociceptive responses in rats (*Ohara et al., 2008*; *Vit et al., 2008*; *Ohara et al., 2008*; *Vit et al., 2008*). Together, these studies suggest that satellite glia are capable of taking up extracellular K$^+$, distributing them throughout a glial syncytium via gap junction coupling, and extruding ions in regions of low K$^+$ concentration, in a process called 'spatial K$^+$ buffering' (*Kuffler, 1967*; *Tang et al., 2010*), similar to astrocyte functions in the CNS (*Kofuji and Newman, 2004*). Even slight elevations in extracellular K$^+$ in the neuronal microenvironment, due to loss of glial Kir4.1, are likely to elicit neuronal depolarization and activation (*Cui et al., 2018*; *Haydon, 2001*). However, Kir4.1 may also function via additional mechanisms to modulate neuronal excitability. For example, Kir4.1 is a major contributor to the hyperpolarized resting membrane potential of satellite glia (*Tang et al., 2010*). Depolarization of satellite glia, induced by loss of Kir4.1, could likely affect voltage-dependent processes in these cells (*Tang et al., 2010*). Kir4.1, expressed in astrocytes, inhibits the synthesis of brain-derived neurotrophic factor (BDNF) in astrocytes (*Kinboshi et al., 2017*; *Ohno et al., 2018*), and astrocyte-derived BDNF has been shown to enhance neuronal activity through presynaptic mechanisms in the ventromedial hypothalamus (*Ameroso et al., 2022*). Thus, an analogous Kir4.1-BDNF pathway might

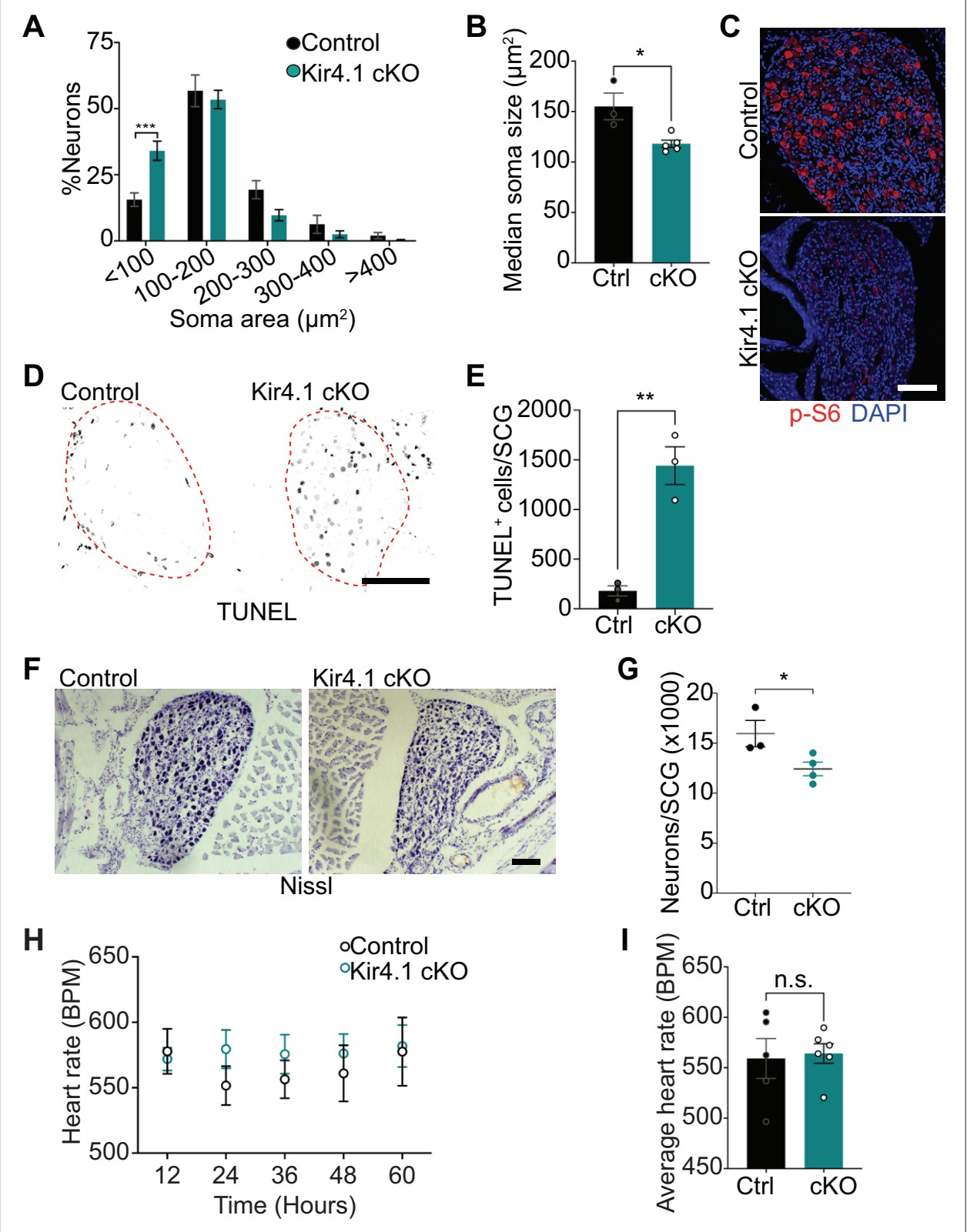

**Figure 5.** Defects in neuron viability in Kir4.1 cKO mice. (**A**) Kir4.1 *cKO* sympathetic neurons have smaller soma sizes compared to control neurons. Results are mean ± SEM from n = 3 control and 5 mutant mice, ***p<0.001, two-way ANOVA with Bonferroni's correction. (**B**) Reduced soma size, represented as median values of soma areas (μm²) of Kir4.1 cKO sympathetic neurons compared to controls. Values are mean ± SEM from n = 3 control and 5 mutant animals, *p<0.05, *t*-test. (**C**) Decreased mTOR signaling based on p-S6 immunostaining in Kir4.1 cKO sympathetic ganglia. Scale

*Figure 5 continued on next page*

*Figure 5 continued*

bar: 100 μm. (**D**) TUNEL labeling shows increased apoptosis in Kir4.1 cKO SCG (outlined in red dashed line). Scale bar: 100 μm. (**E**) Quantification of apoptotic cells in control and mutant sympathetic ganglia from n = 3 mice per genotype. Data presented as mean ± SEM **p<0.01, *t*-test. (**F, G**) Decreased sympathetic neuron numbers in Kir4.1 cKO mice based on Nissl-staining and cell counts in sympathetic ganglia tissue sections. Results are mean ± SEM from n = 3 control and 4 mutant animals. *p<0.05, *t*-test. (**H**) Heart rate is unaffected by Kir4.1 deletion from satellite glia. Results are mean ± SEM from n = 5 control and 6 mutant animals, two-way ANOVA with Bonferroni's correction. (**I**) Average heart rate over days 4–7 post-lead implantation. Results are mean ± SEM from n = 5 control and 6 mutant animals. n.s., not significant.

The online version of this article includes the following source data and figure supplement(s) for figure 5:

**Source data 1.** Raw data for neuron morphology and heart rate in Kir4.1 mutant and control mice.

**Figure supplement 1.** Additional morphology and functional analyses in Kir4.1 cKO mice.

**Figure supplement 1—source data 1.** Raw data for neuron and glia morphology, and pupil analyses in Kir4.1 mutant and control mice.

operate in satellite glia to influence neuron activity since BDNF is known to increase presynaptic input to post-ganglionic sympathetic neurons in sympathetic ganglia (***Causing et al., 1997***).

We found that satellite glia-specific loss of Kir4.1 recapitulates all the cellular phenotypes observed in BLBP:iDTA sympathetic neurons, specifically, increased neuronal activity, diminished mTOR signaling, soma atrophy, downregulated expression of noradrenergic enzymes, and neuronal apoptosis. However, deletion of a single gene, *Kcnj10*, from satellite glia is not sufficient to drive behavioral changes at the whole animal level as accomplished by genetic ablation of these cells. Thus, additional glia-dependent mechanisms must contribute to neuronal excitability to drive network-level changes. Satellite glia could influence neuronal activity via regulation of ganglionic levels of acetylcholine, the major neurotransmitter released by pre-ganglionic axons. Despite previous observations that satellite glia modulate cholinergic neurotransmission in co-cultured sympathetic neurons (***Enes et al., 2020***; ***Feldman-Goriachnik et al., 2018***), we did not detect glial transcripts involved in cholinergic signaling using single-cell RNA-sequencing of sympathetic ganglia (***Mapps et al., 2022***). However, we do not exclude glia-mediated effects on synapse formation during development or maintenance (***Enes et al., 2020***). Another potential mechanism could involve glial regulation of extracellular ATP and/or its breakdown products in the neuronal microenvironment (***Hanani, 2010***). In sympathetic ganglia, ATP is largely released by innervating cholinergic pre-ganglionic axons and facilitates fast excitatory neurotransmission in post-ganglionic sympathetic neurons (***Boehm, 1999***; ***Evans et al., 1992***; ***McCaman and McAfee, 1986***; ***Vizi et al., 1997***). Extracellular ATP is rapidly metabolized by cell-surface ectonucleotidases, which are expressed in sympathetic satellite glia (***Forsman and Elfvin, 1984***; ***Nacimiento et al., 1991***; ***Vizi et al., 1997***). Single-cell RNA-sequencing data revealed several transcripts involved in ATP sensing, hydrolysis, and removal of breakdown products in sympathetic satellite glia, including the purinergic receptors, *P2rx4*, *P2rx7*, and *P2ry12*, and *Entpd1*, an ectoenzyme which catalyzes ATP to ADP hydrolysis, as well as *Adk* (*adenosine kinase*) (***Mapps et al., 2022***), which is best known for mediating astrocytic uptake of extracellular adenosine in the brain (***Boison et al., 2010***).

Our results suggest that autonomic defects observed in BLBP:iDTA mice arise primarily from defects in peripheral sympathetic neurons. We demonstrate that 6-OHDA-mediated ablation of sympathetic nerves prevents elevated heart rate in BLBP:iDTA mice. Since 6-OHDA selectively destroys sympathetic nerves (***Kostrzewa and Jacobowitz, 1974***), these results exclude the involvement of sensory or parasympathetic neurons to the cardiac defect in BLBP:iDTA mice. Parasympathetic function in BLBP:iDTA mice appears to be unaffected since mutant mice fully constrict their pupils in response to light. BLBP is also expressed in neuronal progenitors, radial glia, and astrocytes in the CNS (***Ebrahimi et al., 2016***; ***Feng et al., 1994***; ***Matsumata et al., 2012***). We used *Fabp7-CreER2* mice in an inducible manner in adult age, which allowed us to bypass the targeting of cell types during development. Although we cannot completely exclude an astrocyte contribution to autonomic defects in BLBP:iDTA mice, our results implicate a peripheral locus given the rescue of cardiovascular dysfunction in mutant mice by 6-OHDA, which does not cross the blood–brain barrier in adult mice after intraperitoneal injections (***Kostrzewa and Jacobowitz, 1974***). Our single-cell RNA-sequencing analysis of satellite glia (***Mapps et al., 2022***) compared to published single-cell RNA studies of astrocytes (***Batiuk et al., 2020***) indicates an ~45-fold enrichment of *Fabp7* transcript in satellite glia relative to astrocytes. Further, astrocyte ablation in adult mice results in severe motor deficits, including limb paralysis, ataxia, as well as smaller body weight (***Schreiner et al., 2015***), none of which were observed in BLBP:iDTA mice. Similarly, astrocyte-specific deletion of

*Kcnj10* using *GFAP-cre* mice results in premature lethality, epileptic seizures, and severe ataxia (*Djukic et al., 2007*), none of which were observed with Kir4.1 deletion using *Fabp7-CreER2* mice. Together, these results suggest that astrocytes are minimally perturbed in BLBP:iDTA and Kir4.1 cKO mice.

In a previous study, chemogenetic manipulation of satellite glia by activating a Gq-GPCR signaling pathway increased heart rate in *Gfap-hM3Dq* mice (*Xie et al., 2017*). The experimental manipulation, that is, acute activation (over a time scale of minutes) of a Gq-GPCR pathway in satellite glia using DREADD ligands is different from that in our study (genetic ablation of these cells over 2 weeks), making it challenging to directly compare between the two studies. Also, how activation of a Gq-GPCR signaling pathway and calcium mobilization in satellite glia affects their physiology and function remains unknown. Additionally, we note that cellular specificity of the Cre driver lines used in the two studies is different. We, and others, have shown that *Fabp7* (BLBP) is a specific marker for satellite glia in this study and other work (*Avraham et al., 2022*; *Avraham et al., 2020*; *Avraham et al., 2021*; *Mapps et al., 2022*). Using single-cell sequencing, we, and others, have not detected *Gfap* in mouse satellite glia under normal or reactive conditions (*Jager et al., 2020*; *Mapps et al., 2022*; *Mohr et al., 2021*), although it is found in satellite glia in rats (*Mohr et al., 2021*). Nevertheless, sympathetic satellite glia, very likely, exert both excitatory and inhibitory effects on neuronal activity in a context-dependent manner, similar to astrocyte functions in the brain (*Allen and Lyons, 2018*). Specifically, under conditions of nerve injury or inflammation, activated sensory satellite glia are known to undergo structural and functional changes, resulting in neuronal hyper-excitability (*Hanani and Spray, 2020*). Limited studies, so far, suggest that in response to nerve damage, sympathetic satellite glia are also capable of reactive changes, specifically, enhanced gap junction-mediated coupling and increased ATP sensitivity (*Feldman-Goriachnik and Hanani, 2019*), which, in turn, may augment neuronal activity.

An intriguing finding was that satellite glia depletion or glial deletion of Kir4.1 resulted in the loss of ~25% of adult sympathetic neurons. The exquisite dependence of sympathetic neurons on the target-derived survival factor, nerve growth factor (NGF), during development is well-documented (*Glebova and Ginty, 2005*). However, loss of NGF signaling does not compromise the survival of adult neurons (*Angeletti et al., 1971*; *Tsui-Pierchala and Ginty, 1999*), and to date, the trophic mechanisms underlying adult sympathetic neuron survival in vivo remain undefined. The similar phenotypes of soma atrophy, impaired mTOR signaling, and enhanced apoptosis in adult BLBP:iDTA and Kir4.1 cKO sympathetic neurons suggest that satellite glia provide metabolic and trophic support to mature neurons, primarily, via regulation of $K^+$ homeostasis. Of note, astrocyte-specific Kir4.1 deletion also results in defects in mTOR signaling and decreased soma size, but not cell death, in a population of spinal cord motor neurons (*Kelley et al., 2018*). Whether observed apoptosis of mature sympathetic neurons in BLBP:iDTA or Kir4.1 cKO mice is linked to hyperexcitability of a vulnerable neuronal subpopulation and/or reflects the requirement for glial-derived trophic signals remain to be defined. Further, despite enhanced sympathetic neuron activity and circulating NE levels in satellite glia-ablated mice, we observed a pronounced downregulation of NE biosynthetic machinery, specifically in neuronal cell bodies. This may reflect a compensatory response to elevated sympathetic activity upon glia loss or the need for glia-derived factors in maintenance of noradrenergic enzymes. It is also notable that satellite glia depletion resulted in soma-specific effects in sympathetic neurons, that is, TH downregulation and soma size decrease, without perturbing axonal TH levels and innervation. *Th* mRNA is trafficked to sympathetic axons, where it is locally translated (*Gervasi et al., 2016*). Thus, enhanced *Th* mRNA trafficking and local synthesis might be a mechanism to maintain axonal TH expression and to locally regulate NE release in BLBP:iDTA sympathetic neurons.

Heightened activity of the sympathetic nervous system is a characteristic feature of several pathological conditions, specifically, chronic heart failure (*Malpas, 2010*), arrhythmias (*Hasan, 2013*; *Schwartz, 2014*), hypertension (*Goldstein et al., 2002*; *Schlaich et al., 2004*), sleep apnea, obesity, and insulin resistance (*Mahfoud et al., 2011*; *Mancia et al., 2007*; *Schlaich et al., 2004*). While sympathetic neurons, so far, have been at the center stage in considering pathological mechanisms and treatments, our study, together with other recent work (*Xie et al., 2017*), highlights the therapeutic potential of targeting the neuron–satellite glia unit in autonomic-related diseases.

## Quantification and statistical analyses

Information for statistical analyses for all experiments is provided in 'Materials and methods' and figure legends.

# Materials and methods

### Key resources table

| Reagent type (species) or resource | Designation | Source or reference | Identifiers | Additional information |
|---|---|---|---|---|
| Strain, strain background (*Mus musculus*) | *Fabp7-CreER2* | **Maruoka et al., 2011** | | |
| Strain, strain background (*M. musculus*) | *Gt(ROSA)26Sortm1(DTA)Jpmb/J* | The Jackson Laboratory | RRID:IMSR_JAX:006331 | |
| Strain, strain background (*M. musculus*) | *B6.129-Kcnj10tm1Kdmc/J* | **Djukic et al., 2007** | RRID:IMSR_JAX:026826 | |
| Strain, strain background (*M. musculus*) | *Gt(ROSA)26Sortm4(ACTB-tdTomato,-EGFP)Luo* | **Muzumdar et al., 2007** | RRID:IMSR_JAX:007576 | |
| Strain, strain background (*M. musculus*) | C57Bl/6J | The Jackson Laboratory | RRID:IMSR_JAX:000664 | |
| Antibody | Anti-BLBP (mouse monoclonal) | Abcam | Cat# ab131137 (discontinued); RRID:AB_11157091 | IF (1:500) |
| Antibody | Anti-BLBP (rabbit polyclonal) | Abcam | Cat# ab32423; RRID:AB_880078 | IF (1:200) |
| Antibody | Anti-Sox2 (rabbit polyclonal) | Active Motif | Cat# 39823, RRID:AB_2793356 | IF (1:500) |
| Antibody | Anti-IBA1 (rabbit polyclonal) | WAKO | Cat# 019-19741; RRID:AB_839504 | IF (1:200) |
| Antibody | Anti-Kir4.1 (rabbit polyclonal) | Alomone Labs | Cat# APC-035; RRID:AB_2040120 | IF (1:100) |
| Antibody | Anti-pS6 (rabbit polyclonal) | Cell Signaling | Cat# 2215S; RRID:AB_916156 | IF (1:200) |
| Antibody | Anti-tyrosine hydroxylase (mouse monoclonal) | Sigma | Cat# T2928; RRID:AB_477569 | IF (1:300) |
| Antibody | Anti-tyrosine hydroxylase (rabbit polyclonal) | Millipore | Cat# ab152; RRID:AB_390204 | IF (1:300) |
| Antibody | Anti-c-Fos (rabbit polyclonal) | Abcam | Cat# ab190289; RRID:AB_2737414 | IF (1:1000) |
| Antibody | Anti-Sox10 (goat polyclonal) | R&D Systems | Cat# AF2864; RRID:AB_442208 | IF (1:50) |
| Antibody | Anti-p-4E-BP-1 (rabbit monoclonal) | Cell Signaling | Cat# 2855T; RRID:AB_560835 | IF (1:200) |
| Antibody | Anti-TrkA (rabbit polyclonal) | Millipore | Cat# 06-674; RRID:AB_310180 | IF (1:200) |
| Antibody | Amersham ECL Rabbit IgG, HRP-linked whole Ab from donkey (rabbit polyclonal) | Cytiva | Cat# NA934; RRID:AB_772206 | DAB (1:200) |
| Sequence-based reagent | *Adra1a* TaqMan Probe | Thermo Fisher | Assay ID: Mm00442668_m1 | |
| Sequence-based reagent | *Adra1b* TaqMan Probe | Thermo Fisher | Assay ID: Mm00431685_m1 | |
| Sequence-based reagent | *Adra1d* TaqMan Probe | Thermo Fisher | Assay ID: Mm01328600_m1 | |
| Sequence-based reagent | *Adra2a* TaqMan Probe | Thermo Fisher | Assay ID: Mm00845383_s1 | |
| Sequence-based reagent | *Adra2b* TaqMan Probe | Thermo Fisher | Assay ID: Mm00477390_s1 | |
| Sequence-based reagent | *Adra2c* TaqMan Probe | Thermo Fisher | Assay ID: Mm00431686_s1 | |
| Sequence-based reagent | *Adrb1* TaqMan Probe | Thermo Fisher | Assay ID: Mm00431701_s1 | |
| Sequence-based reagent | *Adrb2* TaqMan Probe | Thermo Fisher | Assay ID: Mm02524224_s1 | |
| Sequence-based reagent | *Adrb3* TaqMan Probe | Thermo Fisher | Assay ID: Mm02601819_g1 | |
| Sequence-based reagent | Eukaryotic *Rn18s* Endogenous Control (VIC/MGB probe, primer limited) | Thermo Fisher | Cat# 4319413E | |

*Continued on next page*

*Continued*

| Reagent type (species) or resource | Designation | Source or reference | Identifiers | Additional information |
|---|---|---|---|---|
| Sequence-based reagent | *Th_F* | This paper | qPCR primers | AATCCACCACTTAGAGACCCG ('Materials and methods') |
| Sequence-based reagent | *Th_R* | This paper | qPCR primers | CTTGGTGACCAGGTGGTGAC ('Materials and methods') |
| Sequence-based reagent | *DBH_F* | This paper | qPCR primers | CATCTGGATTCCCAGCAAGACT ('Materials and methods') |
| Sequence-based reagent | *DBH_R* | This paper | qPCR primers | CAGCGACTGAAATGGCTCTTCC ('Materials and methods') |
| Sequence-based reagent | *Rn18s_F* | **Ceasrine et al., 2018** | qPCR primers | CGCCGCTAGAGGTGAAATTC |
| Sequence-based reagent | *Rn18s _R* | **Ceasrine et al., 2018** | qPCR primers | TTGGCAAATGCTTTCGCTC |
| Sequence-based reagent | *Kcnj10_F* | Harvard Primer Bank | PrimerBank ID:34328498a1 | GTCGGTCGCTAAGGTCTATTACA |
| Sequence-based reagent | *Kcnj10_R* | Harvard Primer Bank | PrimerBank ID:34328498a1 | GGCCGTCTTTCGTGAGGAC |
| Sequence-based reagent | *Fabp7^CreER2 _F* | **Maruoka et al., 2011** | PCR primers | TACCGGTCGACAACGAGTGATGAGG |
| Sequence-based reagent | *Fabp7^CreER2 _R* | **Maruoka et al., 2011** | PCR primers | GACCGACGATGCATGTTTAGCTGG |
| Sequence-based reagent | *Gt(ROSA)26Sortm1(DTA)Jpmb/J _F* | The Jackson Laboratory | PCR primers | AAAGTCGCTCTGAGTTGTTAT |
| Sequence-based reagent | *Gt(ROSA)26Sortm1(DTA)Jpmb/J _R* | The Jackson Laboratory | PCR primers | GCGAAGAGTTTGTCCTCACC |
| Sequence-based reagent | *B6.129-Kcnj10tm1Kdmc/J _F* | The Jackson Laboratory | PCR primers | TGATCTATCTCGATTGCTGC |
| Sequence-based reagent | *B6.129-Kcnj10tm1Kdmc/J _R* | The Jackson Laboratory | PCR primers | CCCTACTCAATGCTCTTAAC |
| Sequence-based reagent | *Gt(ROSA)26Sortm4(ACTB-tdTomato,-EGFP)Luo_WT F* | The Jackson Laboratory | PCR primers | GGC TTA AAG GCT AAC CTG ATG TG |
| Sequence-based reagent | *Gt(ROSA)26Sortm4(ACTB-tdTomato,-EGFP)Luo_WT R* | The Jackson Laboratory | PCR primers | GGA GCG GGA GAA ATG GAT ATG |
| Sequence-based reagent | *Gt(ROSA)26Sortm4(ACTB-tdTomato,-EGFP)Luo_Mut F* | The Jackson Laboratory | PCR primers | CCG GAT TGA TGG TAG TGG TC |
| Sequence-based reagent | *Gt(ROSA)26Sortm4(ACTB-tdTomato,-EGFP)Luo_Mut R* | The Jackson Laboratory | PCR primers | AAT CCA TCT TGT TCA ATG GCC GAT C |
| Chemical compound, drug | Hydrogen peroxide solution, 30% in $H_2O$, ACS reagent | Sigma-Aldrich | Cat# 216763 | |
| Chemical compound, drug | SIGMAFAST 3,3'-Diaminobenzidine tablets, tablet, to prepare 15 mL | Sigma-Aldrich | Cat# D4418-50SET | |
| Chemical compound, drug | 6-Hydrodroxydopamine hydrobromide | Hello Bio | Cat# HB1889 | |
| Chemical compound, drug | Hematoxylin solution | Sigma-Aldrich | Cat# GHS232 | |
| Chemical compound, drug | Bouin's solution | Sigma-Aldrich | Cat# HT10132-1L | |
| Chemical compound, drug | Glycogen | Thermo Fisher | Cat# R0551 | |
| Chemical compound, drug | Maxima SYBR Green/ROX qPCR Master Mix (2×) | Thermo Fisher | Cat# K0222 | |
| Chemical compound, drug | TaqMan Universal PCR Master Mix | Thermo Fisher | Cat# 4304437 | |
| Chemical compound, drug | Trizol | Thermo Fisher | Cat# 15596018 | |
| Other | DAPI | Roche | Cat# 10236276001 | (1 μg/mL) |
| Chemical compound, drug | Agarose, low EEO | Sigma | Cat# A0576-25G | |
| Chemical compound, drug | ProLong Gold Antifade Mountant | Thermo Fisher | Cat# P36930 | |

*Continued on next page*

*Continued*

| Reagent type (species) or resource | Designation | Source or reference | Identifiers | Additional information |
|---|---|---|---|---|
| Chemical compound, drug | Permount Mounting Media | Fisher Scientific | Cat# SP11500 | |
| Chemical compound, drug | Flouromount Mounting Media | Sigma-Aldrich | Cat# F4680-25mL | |
| Commercial assay or kit | Click-iT EdU Alexa Fluor 555 Imaging Kit | Life Technologies | Cat# 10338 | |
| Commercial assay or kit | In Situ Cell Death Detection Kit, TMR red | Roche | Cat# 12156792910 | |
| Commercial assay or kit | Norepinephrine Research ELISA | Rocky Mountain Diagnostics, Inc | BA E-5200 | |
| Commercial assay or kit | Superscript IV First Strand Synthesis System | Thermo Fisher | Cat# 18091050 | |
| Commercial assay or kit | Agilent Absolutely RNA Nanoprep Kit | Agilent | Cat# 400753 | |
| Commercial assay or kit | RNAscope Fluorescent Multiplex Assay | ACD | Cat# 320850 | |
| Commercial assay or kit | RNAscope Probe-Kcnj10-C3 | ACD | Cat# 458831-C3 | |
| Software, algorithm | LabChart 8 software (ADInstruments) | N/A | https://www.adinstruments.com/support/software | |
| Software, algorithm | ImageJ | N/A | https://imagej.nih.gov/ij/ | |
| Software, algorithm | ZEN 2012 SP1 (black edition) | N/A | https://www.zeiss.com/microscopy/int/home.html | |
| Software, algorithm | ZEN 2012 (blue edition) | N/A | https://www.zeiss.com/microscopy/int/home.html | |

## Animal care and housing

All procedures relating to animal care and treatment conformed to The Johns Hopkins University Animal Care and Use Committee (ACUC) and NIH guidelines. Animals were group housed in a standard 12:12 light–dark cycle, except for the pupil analyses. Adult mice between 1 and 1.5 months of age, and of both sexes, were used for analyses. The following mouse lines were used in this study: C57Bl/6J (JAX:000664) mice, *Fabp7-CreER2* mice (**Maruoka et al., 2011**) were generously provided by Dr. Toshihiko Hosoya (RIKEN Brain Science Institute), *Kcnj10$^{fl/fl}$* mice (**Djukic et al., 2007**) (The Jackson Laboratory, stock no: 026826) by Dr. Dwight Bergles (Johns Hopkins School of Medicine), and *ROSA26$^{mEGFP}$* mice (**Muzumdar et al., 2007**) (The Jackson Laboratory, stock no: 007576) by Dr. David Linden (Johns Hopkins School of Medicine). *ROSA26$^{eGFP-DTA}$* mice were obtained from The Jackson Laboratory (stock no: 006331).

## Tamoxifen injections

At postnatal day 30, C57Bl/6J, *Fabp7-CreER2;ROSA26$^{mEGFP}$*, *Fabp7-CreER2;ROSA26$^{eGFP-DTA}$*, or *Fabp7-CreER2;Kcnj10$^{fl/fl}$* mice were injected subcutaneously with either vehicle corn oil (Sigma) or tamoxifen (180 mg/kg body weight) dissolved in corn oil for five consecutive days. All analyses were performed at 5 or 14 days after the last injection.

## Quantitative PCR

RNA was isolated from dissected SCGs or cardiac tissue using Absolutely RNA Nanoprep Kit (Agilent) or Trizol-chloroform extraction. cDNA was prepared using Superscript IV First Strand Synthesis System. Real-time qPCR analysis was performed using Maxima SYBER Green/Rox Q-PCR Master Mix (Thermo Fisher) and gene-specific primers for SCG tissues or TaqMan probes for adrenergic receptors in heart tissue, in a 7300 Real time PCR System (Applied Biosystems). Each sample was analyzed in triplicate reactions. Fold change was calculated using the $2^{(\Delta\Delta Ct)}$ method, normalizing to 18S transcript.

## Single-molecule fluorescence in situ hybridization

SCGs were dissected from P48 mice, cryoprotected in 30% sucrose in PBS for 1 hr and embedded in OCT and frozen at –80°C. Ganglia were cryosectioned at 14 µm and kept at –80°C until smFISH was

performed. Target mRNA was probed using RNAscope Multiplex Fluorescent Reagent Kit v2 Assay. Tissues were incubated in fresh 4% paraformaldehyde for 5 min, washed twice in 1× PBS, and dehydrated with increasing concentrations of ethanol. Subsequently, tissues were treated with hydrogen peroxide for 10 min and protease treatment for 15 min. RNAscope assays were performed following the manufacturer's instructions. Tissues were mounted with Prolong Anti-fade Mounting Media and imaged using a Zeiss LSM 800 confocal microscope.

## Immunohistochemical analyses

SCGs were harvested and incubated in 4% paraformaldehyde for 4 hr at room temperature, cryoprotected in 30% sucrose in 1× PBS for 3 days, embedded in OCT (Sakura Finetek) and stored at –80°C. Ganglia were cryosectioned at 12–30 µm sections for immunohistochemistry. For paraffin embedding, SCGs were fixed in Bouin's solution for 1 hr at room temperature, left in 70% ethanol overnight at room temperature, followed by consecutive washes in 70, 80, 95, and 100% ethanol, and xylene. Tissues were embedded in paraffin wax and sectioned at 6 µm using a microtome, and rehydrated using a series of xylene and ethanol washes at room temperature. Tissue sections were permeabilized in 0.1% Triton X-100 in 1× PBS at room temperature 3× for 5 min each, followed by incubation in 10 mM sodium citrate buffer (pH 6) at 95°C for 10 min. Tissues were blocked in 5% goat serum/3% bovine serum albumin in 0.1% or 0.3% Triton X-100 in 1× PBS (blocking solution) for 1 hr at room temperature. Primary antibodies used were mouse anti-BLBP (1:500), rabbit anti-BLBP (1:200), mouse anti-TH (1:300), rabbit anti-TH (1:300), rabbit anti-TrkA (1:200), rabbit anti-IBA1 (1:200), rabbit anti-pS6 (1:200), rabbit anti-p4E-BP1 (1:200), rabbit anti-c-Fos (1:1000), goat anti-Sox10 (1:50), or rabbit anti-Kir4.1 (1:100) with incubations performed overnight at 4°C. Slides were washed in blocking solution and then incubated in Alexa-488 ,-546, or-647 conjugated anti-rabbit or anti-mouse secondary antibodies (1:200 dilution) and DAPI (1:1000) in blocking buffer. Tissues were then mounted in Aqueous Mounting Medium and imaged at 1 µm optical slices using a Zeiss LSM 700 or LSM 800 confocal microscope. Maximum intensity projections and maximum intensity 3D projections were generated using ImageJ.

For Sox-2 immunostaining, SCGs were embedded in 3% agarose (Sigma) and sectioned at 100-µm-thick sections using a vibratome. Tissues were permeabilized/blocked in 10% goat serum/3% tween in 1× PBS for 2 hr at room temperature, incubated with rabbit anti-Sox2 (1:500) and mouse anti-TH (1:300) antibodies in blocking solution for 2 days at 4°C, followed by Alexa-546 or -647 conjugated anti-rabbit or anti-mouse secondary antibodies (1:400 dilution) and DAPI (1:1000). Maximum intensity 3D projections were generated using ImageJ.

Binary images of Sox2$^+$ cells were generated using Fourier Bandpass Filter plug-in on ImageJ to first reduce edge artifacts, and then adjusting the threshold to the maximum fluorescence intensity of 255 nm. Particles of sizes between 3 and 50 µm were quantified using the 'Analyze Particle' plug-in on ImageJ.

For TH DAB immunohistochemistry, paraffin sections (6 µm) of SCGs were prepared as described above. After tissues were rehydrated, peroxidase quenching was done by incubation in a tris buffer solution (TBS) containing 3% hydrogen peroxide/5% methanol for 15 min at room temperature. Tissues were then permeabilized/blocked in 10% donkey serum/1% glycine/2% bovine serum albumin/0.4% Triton X-100 in 1× TBS for 2 hr at room temperature, incubated with rabbit anti-TH (1:200) in blocking solution overnight at 4°C. Slides were washed three times with TBS, incubated with rabbit anti-HRP in blocking solution for 1 hr at room temperature, washed with TBS before being incubated with SIGMA-FAST 3,3'-diaminobenzidine for 10 min. Tissues were washed once with TBS and counterstained with hematoxylin for 1 min prior to dehydrating in increasing concentrations of ethanol and xylene. Tissues were mounted with Permount Mounting Media. Glial nuclei associated with TH-positive neurons in SCGs were counted using ImageJ.

## iDISCO and wholemount immunostaining

iDISCO-based tissue clearing for wholemount immunostaining of organs from P48 mice was performed as previously described (*Renier et al., 2014*). Briefly, hearts were fixed in 4% PFA/PBS, then dehydrated by methanol series (20–80%) and incubated overnight in 66% dichloromethane (DCM)/33% methanol. Samples were then bleached with 5% H$_2$O$_2$ in methanol at 4°C overnight, then re-hydrated and permeabilized first with 0.2% Triton X-100 followed by overnight permeabilization with 0.16%

Triton X-100/20% DMSO/0.3 M glycine in PBS. Samples were incubated in blocking solution (0.17% Triton X-100/10% DMSO/6% Normal Goat Serum in PBS) for 8 hr, and then incubated with rabbit anti-TH (1:400) in 0.2% Tween-20/0.001% heparin/5% DMSO/3% Normal Goat Serum in PBS at 37°C for 96 hr. Samples were then washed with 0.2% Tween-20/0.001% heparin in PBS and incubated with anti-rabbit Alexa-546 secondary antibody (1:400) in 0.2% Tween-20/0.001% heparin/3% Normal Goat Serum in PBS. After 96 hr, organs were extensively washed with 0.2% Tween-20/0.001% heparin in PBS and dehydrated in methanol. Samples were cleared by successive washes in 66% DCM/33% methanol, 100% DCM, and 100% dibenzyl ether. Organs were imaged on a lightsheet microscope (LaVision BioTec Ultra Microscope II). Imaris was used for 3D manipulations. Total axon lengths and number of branch points were quantified using Imaris Filament Tracer and normalized to total organ volume.

## Soma size

SCG tissue sections were labeled with hematoxylin and eosin. Cells with characteristic neuronal morphology and visible nucleoli were identified, soma were traced and areas ($\mu m^2$) quantified using ImageJ (Fiji).

## Neuronal cell counts

Neuronal counts were performed as previously described (*Scott-Solomon and Kuruvilla, 2020*). In brief, torsos of P39-48 mice were fixed in 4% PFA/PBS overnight and cryoprotected in 30% sucrose/PBS for 48 hr. Torsos were then mounted in OCT and serially sectioned (12 µm). Next, every fifth section was stained with solution containing 0.5% cresyl violet (Nissl). Cells in both SCGs with characteristic neuronal morphology and visible nucleoli were counted using ImageJ.

## TUNEL

Apoptotic cells were identified in every fifth section of ganglia. Tissues were first incubated with primary and secondary antibodies as described above, followed by detection of cell death using TUNEL staining (Roche) according to the manufacturer's protocol. Cells that were double positive for DAPI and TUNEL were counted as dying cells. To quantify neuronal or glial apoptosis, TUNEL labeling was done together with TrkA or Sox10 immunostaining as described above. TUNEL$^+$; TrkA$^+$ and TUNEL$^+$; Sox10$^+$ cells were expressed as percentages of the total number of TrkA$^+$ and Sox 10$^+$ cells, respectively.

## EdU labeling

Tamoxifen- or corn oil-injected adult mice were injected intraperitoneally with EdU (Invitrogen, 100 µg/ml in 3:1 PBS/DMSO) for five consecutive days before harvesting at 5 days post-tamoxifen injections. SCG tissue sections (12 µm) were processed for 30 min at room temperature in EdU reaction cocktail (Thermo Fisher EdU kit C10337; Click-iT buffer, Buffer additive, $CuSO_4$ solution, and Alexa Fluor 488). Sections were then washed in PBS +0.1% Triton X-100 and mounted with Fluoromount + DAPI. Images were collected using a Zeiss LSM 700 confocal microscope. The total number of cells that incorporated EdU in each section was counted and summed for an entire SCG.

## Norepinephrine ELISA

Blood samples drawn from anesthetized mice were centrifuged for 15 min, 3000 rpm in 0.5 M EDTA-coated tubes at 4°C. NE levels in plasma (300 µl) was assessed using a ELISA kit (Abnova) according to the manufacturer's protocol.

## Pupil analyses

Pupil size measurements were performed as reported previously (*Keenan et al., 2016*). Briefly, all mice were dark-adapted and housed in single cages for 2 days and analyzed in the evenings. For all experiments, mice were unanesthetized and restrained by hand. To mitigate stress, which can affect pupil size, researchers handled mice for several days prior to the measurements. Videos of the eye were recorded for 5–10 s in the dark using a Sony Handycam (DCR-HC96) mounted on a tripod at a fixed distance from the mouse. Manual focus was maintained on the camera to ensure that only one

focal plane existed for each mouse. Pupil size was recorded under dim red light and the endogenous infrared light source of the camera to capture the basal pupil size.

To examine parasympathetic activity, mice were dark-adapted for 2 days. Unanesthetized mice were restrained by hand. Pupil size was recorded first for 5–10 s in the dark followed by a 30 s exposure to a light step stimulus. The light stimulus was provided by 10 W or 14 W LED bulbs (Sunlite A19/ or Sunlite 80599-SU LED A19 Super Bright Light Bulb, Daylight). Light intensity was measured using a light meter (EXTECH Foot Candle/Lux Light Meter, 401025) at the surface on which the mouse was held. Light intensity was adjusted by a combination of altering the distance of the light bulb from the mouse and/or applying neutral density filters (Roscolux). The light meter is unreliable at detecting light intensities below 1 lux, so one neutral density filter cutting the light intensity by 12.5% was applied to the bulb to estimate 1-log unit decreases in illumination below 1 lux. Light intensities above 500 lux required the use of multiple light bulbs.

## Electrocardiograms

ECG recordings were performed on adult mice as previously described (*Ståhlberg et al., 2019*). Briefly, adult mice (P39-48) were anesthetized with 4% isoflurane, intubated, and placed on ventilator support (settings 1.2 ml/g/min at 80 breaths/min). The animal's dorsum was shaved, scrubbed with betadine and alcohol, and draped with a sterile barrier with the surgery site exposed. A small 0.5 cm midline incision was performed and ECG leads were implanted subcutaneously and sutured over the trapezius muscle on both sides. Body temperature was maintained at 37°C. Immediately following implantation, the wound was closed with a 3-0 silk suture. Anesthesia was turned off and the animal was monitored for spontaneous breathing and was given a subcutaneous buprenorphine injection (0.01–0.05 mg/kg buprenorphine, IM) to alleviate pain. ECGs were subsequently recorded continuously in conscious animals for approximately 7 days for each mouse using the Powerlab data acquisition device and LabChart 8 software (ADInstruments). Mice were kept at a stable temperature with regular 12 hr light/dark cycle. To exclude the effects of pain and anesthesia, continuous ECG recordings between day 4 and 7 post-lead implantation were only included in the analysis of mean heart rates. HRV was analyzed using LabChart 8 and at 10–12 hr intervals as previously described (*Thireau et al., 2008*).

To assess effects of tamoxifen alone, adult (P39-48) C57Bl/6J or $ROSA26^{eGFP-DTA}$ mice that did not express Cre mice were injected with corn oil or tamoxifen (subcutaneous injections, 180 mg/kg body weight for five consecutive days) and ECGs recorded as described above.

## 6-OHDA injections

For chemical ablation of sympathetic nerves, *Fabp7-CreER2;ROSA26*$^{eGFP-DTA}$ mice were first injected with corn oil or tamoxifen. After 18–20 days, control and BLBP:iDTA mice received two intraperitoneal injections, 3 days apart, of 6-OHDA (150 mg/kg) dissolved in 0.1% ascorbic acid. ECGs were subsequently recorded continuously in conscious animals for approximately 4 days for each mouse as described above.

## Quantification and statistical analyses

Sample sizes in this study were calculated using power analyses using RStudio statistical software. For practical reasons, analyses of innervation, cell death assays, proliferation assays, soma size measurements, and NE ELISA's were performed in a semi-blinded manner. The experimenter was aware of the genotypes but performed each immunostaining and measurements without knowing the genotypes. All physiological experiments (pupil dilation and HRV) were performed in a blinded manner; the experimenter was only aware of the ear tag numbers. All t-tests were performed assuming Gaussian distribution, two-tailed, unpaired, and a confidence interval of 95%. One-way or two-way ANOVA with Bonferroni's correction were performed when more than two groups were compared. All error bars represent the standard error of the mean (SEM).

## Acknowledgements

We thank all members of the Kuruvilla, Zhao, and Hattar labs for helpful comments on the project. This work was supported by NIH R01 awards, NS073751 and NS107342, to RK, DC016065 and

EY027202 to HZ, NIMH intramural research funds (ZIAMH002964) to SH, a NSF GRFP (DGE-1746891) award to AM, and NIH Training grant T32GM007231 to the JHU CMDB graduate program for AM and EB.

## Additional information

### Funding

| Funder | Grant reference number | Author |
|---|---|---|
| National Institutes of Health | NS073751 | Rejji Kuruvilla |
| National Institutes of Health | NS107342 | Rejji Kuruvilla |
| National Science Foundation | DGE-1746891 | Aurelia A Mapps |
| National Institutes of Health | DC016065 | Haiqing Zhao |
| National Institutes of Health | EY027202 | Haiqing Zhao |
| National Institutes of Health | ZIAMH002964 | Samer Hattar |

The funders had no role in study design, data collection and interpretation, or the decision to submit the work for publication.

### Author contributions

Aurelia A Mapps, Conceptualization, Data curation, Formal analysis, Investigation, Methodology, Writing - original draft, Writing – review and editing; Erica Boehm, Corinne Beier, Jennifer Langel, Methodology; William T Keenan, Methodology, Writing – review and editing; Michael Liu, Investigation; Michael B Thomsen, Resources; Samer Hattar, Resources, Funding acquisition, Writing – review and editing; Haiqing Zhao, Resources, Funding acquisition, Project administration, Writing – review and editing; Emmanouil Tampakakis, Investigation, Methodology, Writing – review and editing; Rejji Kuruvilla, Conceptualization, Resources, Funding acquisition, Writing - original draft, Project administration, Writing – review and editing

### Author ORCIDs

Aurelia A Mapps  http://orcid.org/0000-0002-7956-2465
Corinne Beier  http://orcid.org/0000-0002-0698-7219
Samer Hattar  http://orcid.org/0000-0002-3124-9525
Haiqing Zhao  http://orcid.org/0000-0003-4275-9843
Rejji Kuruvilla  http://orcid.org/0000-0002-2851-675X

### Ethics

All procedures relating to animal care and treatment conformed to The Johns Hopkins University Animal Care and Use Committee (ACUC, protocol#MO19A488) and NIH guidelines.

### Decision letter and Author response

Decision letter https://doi.org/10.7554/eLife.74295.sa1
Author response https://doi.org/10.7554/eLife.74295.sa2

## Additional files

### Supplementary files
• Transparent reporting form

## Data availability

All data generated or analysed during this study are included in the manuscript (Results, Materials and Methods, and Figure Legends).

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
