## [Editor Report]

This study covers the understudied topic of satellite glia and its impact on sympathetic neuron survival. An unexpected increase in neuronal activity was observed when satellite glia were deleted by mouse genetics. Novel roles of satellite glia in sympathetic physiology were uncovered by physiology assays. An extensive response to the reviewers, along with the addition of supporting data, was provided by the authors.

---

## [Decision Letter]

**Decision letter after peer review:**

Thank you for submitting your article "Satellite glia are essential modulators of sympathetic neuron survival, activity, and autonomic function" for consideration by *eLife*. Your article has been reviewed by 3 peer reviewers, and the evaluation has been overseen by a Reviewing Editor and Ma-Li Wong as the Senior Editor. In addition, the following individual involved in the review of your submission has agreed to reveal their identity: Bruce D Carter (Reviewer #3).

The reviewers have discussed their reviews with one another, and the Reviewing Editor has drafted this letter to help you prepare a revised submission. There is general agreement that the subject of satellite glial cells (SCG) is of considerable interest. However, the cellular ablation experiments brought several inconsistencies and matters of interpretation, which involve the impact of SGC upon sympathetic function.

Essential revisions:

1) The two targeted ablations with brain lipid binding protein (Blbp) directing diphtheria toxin and deletion of Kir4.1 require additional support for the conclusion that satellite glia are directly involved in the cellular effects on sympathetic and autonomic function. Since opposite actions are detected, it is necessary to reconcile the impact of satellite glia-ablation mice. This concern also raises questions about the specificity and efficacy of targeted expression.

2) It should be determined whether Kir4.1 is present in other cell types. Also, whether Blbp is driving diphtheria toxin in different cells. The effects of the Kir4.1 deletion may be due to alternative explanations, besides altered K buffering.

3) There is also a need to show that SGCs are depleted from sympathetic ganglia.

4) The results of the TUNEL assays require clarification and should be verified with glial and neuronal markers.

Extensive comments from the three reviewers are included below.*Reviewer #1 (Recommendations for the authors):*

There are numerous concerns regarding result interpretation. Therefore, the comments below address written text and figures.

Specific comments:

Title: "Essential modulators" is a subjective evaluation based on a single manipulation that may cause effects other than the removal of satellite glia. Suggested title: "Satellite glial cells modulate sympathetic neuron survival, activity, and autonomic function."

l.28. The term "how" is ambiguous. Suggest replacing with "the extent to which" or "whether," unless the intent is to determine the mechanism, which this study does not.

l.35 and l.81. Data do not so clearly support these statements. Yes, neuronal cellular deficits are recapitulated by Kir4.1 deletion in SGCs, but autonomic functional alterations are not, and the effects of Kir4.1 deletion are not proven to result from altered K buffering.

l.51. Satellite glia in parasympathetic and sensory ganglia have similar "location and morphology"; please clarify the distinction that is meant.

Figure 1 B. Staining in control for Blbp:iDTA appears more similar to p14 than p31 staining in panels in A. Moreover, in the Blbp:iDTA section, are SGCs less intense, thinner, or discontinuous around neurons? Scale bars in B,C, and upper panel in E are stated as 100 um. In comparison, cell diameters are clearly larger in E. Please verify calibrations. TUNEL labeling in C does not seem representative of quantification in D (minimal apparent labeling in control, vast amount in DTX); suggest black and white image without DAPI. Is there a difference in DAPI+ cells?

l.135. The data show that DTX deletion results in NE enzyme depletion; whether that reflects a necessity for SGCs to maintain the enzymes or DTX triggers their decline is not examined here.

L141. Soma areas should be compared using geometric means (median values) because data distribution is non-Gaussian.

l.145 and l.238. The use of pS6 to define mTOR activity is not very conclusive; others use it to measure neuronal activity, for example.

l. 172. Eye diameters differ in Figure 3A; please replace them with comparable images and add a scale bar.

l.199. The cFOS labeling and quantification appear at odds with each other. Staining in F seems to show many cells (even more than in Kir4.1 illustration Figure 4E), yet a mean 100 is shown in G. There are about 12000 neurons/ganglion (Figure 2F) so that fewer than 1% should be positive. Please illustrate with representative images.

l.218. There is concern that Kir4.1 appears to be present in some neurons in Figure 4A (upper left quadrant); please check this.

l.221. What is meant by SGC "overall morphology"? Overall, Blbp staining and complexity of SGCs seem less in the cKO in Figure S4B.

l.229. Was TH staining normal in axons of cKO mice?

l.230. Please change "indicate" for a such as "suggest" or a phrase such as "are consistent with."

l.236. Please provide median soma areas.

l. 281. Spatial buffering should be considered in more detail, as indicated in public comments.

l.319. The opposite conclusions reached in a previous study using a different approach should be discussed in more detail.

l.329. Trophic effects previously reported for Kir4.1 include BDNF production and glutamate buffering; please expand the discussion to indicate roles of Kir4.1 in addition to K buffering.

*Reviewer #2 (Recommendations for the authors):*

– Essential has a specific genetic meaning (i.e., when an essential-for-a-process gene is lacking, the process does not occur). The use of "essential" throughout the manuscript, including the title, to refer to SGCs' role is therefore not formally correct and may be misleading to the reader.

– The authors quantify *Sox2*(+) cells not associated with neuronal cell bodies, but since *Sox2* is a nuclear marker, how can the authors be sure that the cytoplasm associated with the quantified *Sox2*(+) nuclei are not in contact with neuronal cell bodies?

– The link between mTOR and SGC ablation is somewhat disjointed. Can the authors clarify in the text what the mechanistic link they envision is?

– In all figure legends, it would be helpful for the reader to indicate the time-point being examined. The absence of this information made it difficult to interpret the data mentioned above in Figure 1E vs. 2A, for example.

*Reviewer #3 (Recommendations for the authors):*

Previous studies have indicated that satellite glia around sensory neurons in the DRG are also important for phagocytosing apoptotic neurons during development (Wu et al., 2009, PMC2834222). It would be interesting to know if satellite glia have a similar phagocytic role in the sympathetic ganglia, even in adult mice. Did the authors observe any of the apoptotic cells being engulfed by glia vs. macrophages? This is not necessary to address in this manuscript; I was just curious.

---

## [Author Response]

Essential revisions:1) The two targeted ablations with brain lipid binding protein (Blbp) directing diphtheria toxin and deletion of Kir4.1 require additional support for the conclusion that satellite glia are directly involved in the cellular effects on sympathetic and autonomic function. Since opposite actions are detected, it is necessary to reconcile the impact of satellite glia-ablation mice. This concern also raises questions about the specificity and efficacy of targeted expression.

We have performed additional experiments and analyses to demonstrate the specificity of BLBP and Kir4.1 expression in satellite glial cells. We also include new results that strengthen our conclusions that deficits in sympathetic neurons are responsible for autonomic dysfunction in satellite glia-ablated mice (BLBP:iDTA mice).

(i) We analyzed the expression of BLBP and Kir4.1 in peripheral ganglia using data-sets from our recently published single-cell RNA sequencing analysis of sympathetic and sensory ganglia from adult mice (postnatal days 30-45) (Mapps et. al, *Cell Rep* 2022). We show that BLBP (*Fabp7*) and Kir4.1 (*Kcnj10*) are highly enriched in satellite glial cells compared to other ganglionic cell types (∼140-fold enrichment for *Fabp7* and ∼3-fold for *Kcnj10*, respectively). These additional analyses are included in Figure 1—figure supplement 1A and Figure 4—figure supplement 1A, revised manuscript.

(ii) To further ensure the cellular specificity of BLBP, we generated genetic reporter mice by crossing *Fabp7-CreER2* mice with *ROSA26^mEGFP^* mice, which drives expression of membrane-tagged EGFP. We observed m-EGFP reporter expression in satellite glia, but not in neurons, macrophages, or vascular mural cells in sympathetic ganglia, as revealed by co-immunolabeling studies. These new results are included in Figure 1—figure supplement 1C-F in the revised manuscript.

Together, these results indicate that BLBP and Kir4.1 are satellite glia-specific molecular markers. Our results are consistent with published data from other groups using single cell-sequencing of satellite glia (Avraham *et al.*, 2020; 2021; 2022), analyses of *Fabp7-CreER2* reporter mice (Avraham *et al.*, 2020), and Kir4.1 immunolabeling in peripheral ganglia (Vit *et al.*, 2006; 2008).

(iii) Reviewer #1 asked if the autonomic dysfunction observed in satellite glia-ablated mice (BLBP:iDTA mice) could be attributed to involvement of other cell types, for example, due to perturbed sensory or parasympathetic activity. In the original submission, we provided evidence that light-induced pupil constriction, a readout of parasympathetic function, is normal in BLBP:iDTA mice. To further strengthen the involvement of sympathetic neurons in the behavioral defects observed in BLBP:iDTA mice, we ablated sympathetic nerves using 6-OHDA, which selectively destroys sympathetic, but not parasympathetic or sensory nerves (Kostrzewa and Jacobowitz, 1974; Borden *et al.*, 2013). Further, 6-OHDA does not cross the blood-brain barrier after intraperitoneal injections in adult animals (Kostrzewa and Jacobowitz, 1974), restricting its actions to the periphery.

We include new data to show that ablation of sympathetic nerves prevents the elevation in heart rate in BLBP:iDTA mice (Figures 3E-F, revised manuscript). These results support the conclusion that the elevated heart rate in BLBP:iDTA mice specifically arises from defects in peripheral sympathetic neurons.

(iv) We respectfully disagree with the comment that “*opposite actions are detected*” in BLBP:iDTA and Kir4.1 cKO mice. Loss of Kir4.1 in satellite glia recapitulates all cellular phenotypes observed in BLBP:iDTA sympathetic neurons, specifically, increased neuronal activity, diminished mTOR signaling, soma atrophy, down-regulated expression of noradrenergic enzymes, and neuronal apoptosis. However, deletion of a single gene, Kir4.1, from satellite glia is not sufficient to drive behavioral changes at the whole animal level, as accomplished by genetic ablation of satellite glial cells. A reasonable explanation is that there are other satellite glia mechanisms, in addition to Kir4.1 activity, that contribute to regulation of neuronal excitability to drive circuit-level changes. We have discussed these additional mechanisms, for example, regulation of ATP signaling, in the Discussion (pages 16-17, revised manuscript).

2) It should be determined whether Kir4.1 is present in other cell types. Also, whether Blbp is driving diphtheria toxin in different cells.

See response above to point #1 regarding additional experiments and analyses that we performed to address cellular specificity of Kir4.1 and BLBP in satellite glia.

We also note that murine cells are naturally resistant to the uptake of diphtheria toxin (DT) because they lack receptors that bind to diphtheria toxin-subunit B (DTB), which is needed for the cellular entry of diphtheria subunit A (DTA) that has the catalytic activity for inhibition of protein synthesis in cells (Naglich et al., 1992). The *ROSA26^eGFP-DTA^* mice has been used in several studies in combination with specific Cre-driver lines, and to our knowledge, there are few reports of non-cell-autonomous ablation. Thus, we conclude that in BLBP:iDTA mice, DTA expression is restricted to BLBP-positive satellite glial cells.

The effects of the Kir4.1 deletion may be due to alternative explanations, besides altered K buffering.

Reviewer #1 asks an interesting question of whether the sympathetic neuron phenotypes observed in Kir4.1 cKO mice could be due to mechanisms other than the reported function of Kir4.1 in spatial K^+^ buffering. The Reviewer also suggests that peripheral satellite glia may not play a predominant role in K^+^ buffering as their astrocytic counterparts in the CNS. We note that electrophysiological analyses show that satellite glial cells have high K^+^ conductance, which is almost exclusively dependent on Kir4.1 expression (Tang *et al.*, 2010). Satellite glia are coupled to one another through gap junctions (Huang *et al.*, 2005; Kim *et al.*, 2016). RNAi-mediated knockdown of Kir4.1 or Connexin 43, a gap junction protein, enhances excitability of sensory neurons and evokes nociceptive responses in rats (Ohara *et al.*, 2008; Vit *et al.*, 2008). In this study, we find that satellite glia-specific Kir4.1 deletion result in increased sympathetic neuron activity, as revealed by enhanced c-Fos expression.

Together, these results suggest that satellite glial cells are capable of influencing neuronal excitability by dissipating extracellular K^+^ increases via flow through a syncytium of coupled glial cells with high K^+^ conductance, similar to astrocytes in the CNS (Tang *et al.*, 2010).

However, we agree with the Reviewer that we cannot exclude additional functions of Kir4.1. We have toned down the emphasis on spatial K^+^ buffering as the primary mechanism by which Kir4.1 affects sympathetic neuron excitability and have expanded the Discussion (pages 15-16, revised manuscript) to discuss additional mechanisms, for example, the contribution of Kir4.1 to the hyperpolarized resting membrane potential of satellite glial cells (Tang et. al, 2010). Satellite glia membrane depolarization due to the loss of Kir4.1 could likely influence voltage-dependent processes in these cells. It has also been reported that loss of Kir4.1 enhances astrocytic BDNF release (Kinboshi *et al.*, 2017; Ohno *et al.*, 2018), which could then influence neuronal activity by regulating presynaptic mechanisms (Ameroso *et al.,* 2022). Thus, an analogous Kir4.1-BDNF pathway might operate in satellite glia to influence neuron activity, since BDNF is known to increase pre-synaptic input to post-ganglionic sympathetic neurons in sympathetic ganglia (Causing et al., 1997).

3) There is also a need to show that SGCs are depleted from sympathetic ganglia.

We include new data to show the ablation of satellite glial cells in BLBP:iDTA sympathetic ganglia.

(i) We performed co-labeling for TUNEL and Sox10, a satellite glia marker, to show enhanced apoptosis of satellite glial cells at 5 days post-tamoxifen injection in BLBP:iDTA sympathetic ganglia (Figures 1E, G, revised manuscript).

(ii) By q-PCR analyses, we show that levels of several satellite glia-specific transcripts, including *Fabp7* and *Apoe*, are significantly reduced in BLBP:iDTA sympathetic ganglia (Figure 1—figure supplement 2H, revised manuscript).

(iii) We directly addressed Reviewer #2’s comment that “the sox2^+^ satellite glial cells do not appear to be gone 14 days post-tamoxifen injection, just dimmer” in Figure 1E of the original submission. We generated binary images of the Sox2-labeled cells by filtering and thresholding using Image J (details are provided in the Methods, page 40, revised manuscript). This method allows us to simply record the presence or absence of cells in the images in a manner independent of pixel values. Quantification of binary images revealed a significant decrease in the number of Sox2-labeled cells (33% decrease) in BLBP:iDTA ganglia at 14 days post-tamoxifen injections compared to controls (Figure 1—figure supplement 2D-G, revised manuscript). The 33% decrease in satellite glial cells is lower than the 54% loss we had initially reported, suggesting that we may have initially included some proportion of cells that had down-regulated Sox2 expression, but had not been ablated. We have clarified this point in the Results (page 7, revised manuscript).

As an additional measure to visualize sympathetic neurons and associated satellite glial cells in a manner that does not rely on fluorescence, we labeled sympathetic neurons using Tyrosine Hydroxylase (TH) DAB (3,3'-diaminobenzidine) immunohistochemistry, and identified the glial nuclei surrounding TH-positive sympathetic neurons by their distinctive appearance and location using Hematoxylin staining. Of note, despite several attempts, we were not successful in using *Sox2* or Sox10 antibodies in DAB immunohistochemistry. TH DAB immunohistochemistry and Hematoxylin labeling revealed a significant loss of glial nuclei associated with the soma of individual sympathetic neurons in BLBP:iDTA mice compared to control mice, at 14 days post-tamoxifen injection (Figures 1K-L, revised manuscript).

Together, with our data in the original submission, these new results demonstrate that we are indeed successfully ablating satellite glial cells in BLBP:iDTA mice, although the ablation is not complete.

4) The results of the TUNEL assays require clarification and should be verified with glial and neuronal markers.

We performed TUNEL labeling together with immunostaining for TrkA, a sympathetic neuron marker, and Sox10, a satellite glial cell marker, in BLBP:iDTA mice and control litter-mates at 5 days post-tamoxifen injection. We show that the majority of apoptotic cells at this stage are satellite glial cells; we quantified a 2-fold increase in glial apoptosis in BLBP:iDTA sympathetic ganglia at 5 days post-tamoxifen injection. Although, there was a trend toward enhanced neuronal apoptosis at this early stage, the number of TUNEL^+^ sympathetic neurons in BLBP:iDTA ganglia was not statistically different from that in controls. These new results, included in Figures 1E-H, revised manuscript, suggest that satellite glia apoptosis precedes neuronal apoptosis in BLBP:iDTA sympathetic ganglia

Similar co-labeling analyses conducted in Kir4.1 cKO and control mice 14 days post-tamoxifen injections indicate that satellite glia-specific Kir4.1deletion elicits neuronal, but not, glial apoptosis, consistent with what we initially reported in the original submission. These new results are included in Figure 5—figure supplement 1B-E, revised manuscript.

Extensive comments from the three reviewers are included below.Reviewer #1 (Recommendations for the authors):There are numerous concerns regarding result interpretation. Therefore, the comments below address written text and figures.Specific comments:Title: "Essential modulators" is a subjective evaluation based on a single manipulation that may cause effects other than the removal of satellite glia. Suggested title: "Satellite glial cells modulate sympathetic neuron survival, activity, and autonomic function."

We have changed the title as suggested by the Reviewer.

l.28. The term "how" is ambiguous. Suggest replacing with "the extent to which" or "whether," unless the intent is to determine the mechanism, which this study does not.

As suggested, we have modified this sentence to “.., the extent to which sympathetic functions are influenced by satellite glia in vivo remains unclear”

l.35 and l.81. Data do not so clearly support these statements. Yes, neuronal cellular deficits are recapitulated by Kir4.1 deletion in SGCs, but autonomic functional alterations are not, and the effects of Kir4.1 deletion are not proven to result from altered K buffering.

We have modified the sentences to remove the emphasis on K^+^ buffering.

l.51. Satellite glia in parasympathetic and sensory ganglia have similar "location and morphology"; please clarify the distinction that is meant.

We have modified this sentence to “Satellite glia have been largely characterized by their distinctive location and morphology in peripheral ganglia”.

Figure 1 B. Staining in control for Blbp:iDTA appears more similar to p14 than p31 staining in panels in A.

We find that the pattern of thin satellite glial covers enveloping neurons in P31 day-old wild-type mice in Figure 1A is consistent with the overall satellite glia morphology observed in control animals of similar age in Figure 1B. We also note that the magnification is different between the two images.

Moreover, in the Blbp:iDTA section, are SGCs less intense, thinner, or discontinuous around neurons?

Yes, this is consistent with satellite glia depletion in BLBP:iDTA sympathetic ganglia.

Scale bars in B,C, and upper panel in E are stated as 100 um. In comparison, cell diameters are clearly larger in E. Please verify calibrations.

We thank the Reviewer for pointing this out. We have corrected the scale bars for these images.

TUNEL labeling in C does not seem representative of quantification in D (minimal apparent labeling in control, vast amount in DTX); suggest black and white image without DAPI. Is there a difference in DAPI+ cells?

The graph in Figure 1D represents the total number of TUNEL-positive cells in the entire superior cervical ganglia (approximately 24-32 tissue sections of 12 μm thickness each), while images in Figure 1C show a single tissue section from the ganglia. We have inverted the images for better visualization as suggested.

There are no significant differences in the total number of sympathetic neurons between BLBP:iDTA and control ganglia at 5 days post-tamoxifen injections, when the TUNEL labeling was performed. These new results are provided in Figure 2—figure supplement 1B in the revised manuscript.

l.135. The data show that DTX deletion results in NE enzyme depletion; whether that reflects a necessity for SGCs to maintain the enzymes or DTX triggers their decline is not examined here.

As noted above, DTA is only expressed in BLBP-positive satellite glial cells. Hence, we conclude that decreased expression of Norepinephrine (NE) biosynthetic enzymes in sympathetic neurons is an indirect effect of satellite glia loss.

L141. Soma areas should be compared using geometric means (median values) because data distribution is non-Gaussian.

Done. The new analyses are shown for BLBP:iDTA and Kir4.1 cKO mice in Figures 2D and 5B, respectively, in the revised manuscript.

l.145 and l.238. The use of pS6 to define mTOR activity is not very conclusive; others use it to measure neuronal activity, for example.

Phospho-S6 (p-S6) is commonly used as a readout of mTOR activity (Meyuhas et. al, 2008). In the revised manuscript, we show that phosphorylation of another well-established downstream effector of mTOR activity, phospho-4E-BP1 (p-4E-BP1) (Saxton and Sabatini, 2017), is also reduced in BLBP:iDTA sympathetic neurons (Figure 2—figure supplement 1A) and Kir4.1 cKO sympathetic neurons (Figure 5—figure supplement 1A). Together, these results strengthen our initial conclusions that mTOR activity is reduced in sympathetic neurons with satellite glia ablation or satellite glia-specific Kir4.1 deletion.

Although p-S6 is also used as a readout for neuronal activity (Knight et al., 2012), this may not be the case for all neuronal populations. We observed enhanced sympathetic neuron activity in BLBP:iDTA and Kir4.1 cKO mice, based on enhanced c-Fos expression although these neurons had reduced p-S6 immunoreactivity.

l. 172. Eye diameters differ in Figure 3A; please replace them with comparable images and add a scale bar.

The eye diameters are different because BLBP:iDTA mice have increased sympathetic tone, resulting in larger pupil areas. We have added a scale bar to the images.

l.199. The cFOS labeling and quantification appear at odds with each other. Staining in F seems to show many cells (even more than in Kir4.1 illustration Figure 4E), yet a mean 100 is shown in G. There are about 12000 neurons/ganglion (Figure 2F) so that fewer than 1% should be positive. Please illustrate with representative images.

We re-analyzed the c-Fos data and have presented them as the percentage of TH-positive sympathetic neurons that are c-Fos-positive in the revised manuscript. We show that 21% of sympathetic neurons are c-Fos positive in BLBP:iDTA ganglia compared to 2.5% in controls (Figure 3I, revised manuscript). Similarly, 28% of sympathetic neurons are c-Fos-positive in Kir4.1 cKO ganglia compared to 2.4% in control ganglia (Figure 4F, revised manuscript).

l.218. There is concern that Kir4.1 appears to be present in some neurons in Figure 4A (upper left quadrant); please check this.

Kir4.1 (*Kcnj10* mRNA) is barely detected in sympathetic neurons based on single-cell RNA sequencing and immunostaining analyses (Figure 4—figure supplement 1A and Figure 1—figure supplement 1C). We replaced the original image in Figure 4A, revised manuscript.

l.221. What is meant by SGC "overall morphology"? Overall, Blbp staining and complexity of SGCs seem less in the cKO in Figure S4B.

We performed qPCR analysis to show that BLBP expression is unaffected in Kir4.1 cKO ganglia (Figure 4—figure supplement 1E)**.** We have also replaced the original image with one that shows that satellite glia morphology is comparable between Kir4.1 cKO and control sympathetic ganglia, based on BLBP immunostaining, (Figure 4—figure supplement 1F).

l.229. Was TH staining normal in axons of cKO mice?

TH staining in Kir4.1 cKO axons appears to be similar to that in control mice. We have included TH immunostaining and quantification of innervation density in new supplemental data (Figure 5—figure supplement 1F-G, revised manuscript).

l.230. Please change "indicate" for a such as "suggest" or a phrase such as "are consistent with."

Done

l.236. Please provide median soma areas.

Done

l. 281. Spatial buffering should be considered in more detail, as indicated in public comments.

See detailed response to this point above in “Essential Revisions” and Reviewer 1’s “Public Review”.

l.319. The opposite conclusions reached in a previous study using a different approach should be discussed in more detail.

See detailed response above to the comments regarding the Xie *et al.*, study.

l.329. Trophic effects previously reported for Kir4.1 include BDNF production and glutamate buffering; please expand the discussion to indicate roles of Kir4.1 in addition to K buffering.

See detailed response to this point above in “Essential Revisions” and Reviewer #1’s “Public Review”.

Reviewer #2 (Recommendations for the authors):– Essential has a specific genetic meaning (i.e., when an essential-for-a-process gene is lacking, the process does not occur). The use of "essential" throughout the manuscript, including the title, to refer to SGCs' role is therefore not formally correct and may be misleading to the reader.

This has been corrected in the Title and elsewhere in the text as suggested.

– The authors quantify Sox2(+) cells not associated with neuronal cell bodies, but since Sox2 is a nuclear marker, how can the authors be sure that the cytoplasm associated with the quantified Sox2(+) nuclei are not in contact with neuronal cell bodies?

The Reviewer raises a good point. We cannot completely conclude whether satellite glial cells attach or not to neuronal soma based on labeling for *Sox2*, a nuclear marker. However, under normal conditions, given the close contacts between satellite glial cells and neuronal soma, with ultra-structural studies showing a 20 nm separation between neuronal and glial membranes (Pannese, 1981), satellite glia nuclei tend to have a characteristic arrangement often in a ring-like pattern around cell bodies of individual neurons. We only quantified *sox2*^+^ nuclei that were found in such an arrangement, immediately juxtaposed to neuronal soma that were delineated by TH immunolabeling.

– The link between mTOR and SGC ablation is somewhat disjointed. Can the authors clarify in the text what the mechanistic link they envision is?

Struck by the pronounced soma atrophy in sympathetic neurons from BLBP:iDTA and Kir4.1 cKO mice, we assessed mTOR signaling given its known role in regulating soma size and neuronal metabolism (Kwon et al., 2006; van Diepen et al., 2009; Zhou et al., 2009). We found a marked decrease in expression of p-S6, a key downstream effector of mTOR, using immunostaining. Notably, other groups have also reported that astrocyte-specific loss of Kir4.1 compromises mTOR signaling in spinal cord motor neurons leading to soma atrophy (Kelly et. al, 2018). In our study, to further support reduced neuronal mTOR signaling after satellite glia ablation or satellite glia-specific loss of Kir4.1, we include new results that show significant decrease in p-4-EBP1, another well-established readout of mTOR activity in BLBP:iDTA and Kir4.1 cKO sympathetic neurons (Figure 2—figure supplement 1A and Figure 5—figure supplement 1A).

These results suggest that glial cells (satellite glial cells and astrocytes) provide trophic support to neurons and promote soma growth, in part, via regulating mTOR signaling in neurons. We have clarified this point in the revised text (page 9).

– In all figure legends, it would be helpful for the reader to indicate the time-point being examined. The absence of this information made it difficult to interpret the data mentioned above in Figure 1E vs. 2A, for example.

Done.

Reviewer #3 (Recommendations for the authors):Previous studies have indicated that satellite glia around sensory neurons in the DRG are also important for phagocytosing apoptotic neurons during development (Wu et al., 2009, PMC2834222). It would be interesting to know if satellite glia have a similar phagocytic role in the sympathetic ganglia, even in adult mice. Did the authors observe any of the apoptotic cells being engulfed by glia vs. macrophages? This is not necessary to address in this manuscript; I was just curious.

This is an excellent point. As noted above to a similar point by Reviewer #2, it remains unknown whether satellite or macrophages or both are involved in clearing apoptotic debris in sympathetic ganglia. Although addressing this question is beyond the scope of this study, we plan to examine the phagocytic properties of satellite glia in future studies through sparse genetic labeling and confocal imaging and/or by electron microscopy.